# Identifying immune signatures of sepsis to increase diagnostic accuracy in very preterm babies

A. Das [1,9] ✉, G. Ariyakumar[1], N. Gupta[2,3], S. Kamdar[1], A. Barugahare [4], D. Deveson-Lucas[4], S. Gee[1], K. Costeloe[5], M. S. Davey [6,7], P. Fleming[5,8,10] & D. L. Gibbons [1,10] ✉

Bacterial infections are a major cause of mortality in preterm babies, yet our understanding of early-life disease-associated immune dysregulation remains limited. Here, we combine multi-parameter flow cytometry, single-cell RNA sequencing and plasma analysis to longitudinally profile blood from very preterm babies (<32 weeks gestation) across episodes of invasive bacterial infection (sepsis). We identify a dynamically changing blood immune signature of sepsis, including lymphopenia, reduced dendritic cell frequencies and myeloid cell HLA-DR expression, which characterizes sepsis even when the common clinical marker of inflammation, C-reactive protein, is not elevated. Furthermore, single-cell RNA sequencing identifies upregulation of amphiregulin in leukocyte populations during sepsis, which we validate as a plasma analyte that correlates with clinical signs of disease, even when C-reactive protein is normal. This study provides insights into immune pathways associated with early-life sepsis and identifies immune analytes as potential diagnostic adjuncts to standard tests to guide targeted antibiotic prescribing.

Rates of late onset sepsis (LOS; sepsis occurring more than seventy-two hours after birth) are highest in preterm, very low birthweight babies (<1500 g)[1,2]. Within this vulnerable cohort, not only is LOS amongst the leading causes of death[3], it also can confer significant life-long morbidity on survivors via association with long-term neurodevelopmental disability and higher rates of cerebral palsy[4–6].

Sepsis does not have a consensus definition in babies less than twenty-eight days old, though it frequently refers to bacterial infection confirmed by positive cultures from blood or cerebrospinal fluid (herein, we refer to this as Microbiologically Confirmed Sepsis; MCS)[7,8].

However, in recognition that few suspected cases meet this criterion and that fatal bacterial infection in preterm babies has been proven at autopsy without pre-mortem positive blood cultures[9], a sub-cohort of cases who nevertheless fulfil clinical or laboratory criteria for invasive infection, are recognized as a separate 'Clinical Sepsis (ClinSep)' group[1,10].

Mechanisms that underpin immune dysregulation within these sepsis groups in early life are poorly understood. This relates not only to technical challenges of obtaining granular information from low volume blood samples, but also to uncoupling disease-associated

[1]Peter Gorer Department of Immunobiology, School of Immunology and Microbial Sciences, King's College London, Guy's Hospital, London, UK. [2]Department of Neonatology, Evelina London Neonatal Unit, Guy's and St Thomas' NHS Foundation Trust, London, UK. [3]Faculty of Life Sciences & Medicine, King's College London, London, UK. [4]Bioinformatics Platform and Department of Microbiology, Biomedicine Discovery Institute, Monash University, Clayton, VIC 3800, Australia. [5]Barts and the London School of Medicine and Dentistry, Queen Mary University of London, London, UK. [6]Infection and Immunity Program and Department of Biochemistry and Molecular Biology, Biomedicine Discovery Institute, Monash University, Clayton, VIC 3800, Australia. [7]Division of Biomedical Sciences, Warwick Medical School, University of Warwick, Coventry CV4 7AL, UK. [8]Department of Neonatology, Homerton Healthcare NHS Foundation Trust, London, UK. [9]Present address: Division of Infection and Immunity, University College London, London WC1E 6BT, UK. [10]These authors contributed equally: P. Fleming, D. L. Gibbons. ✉e-mail: abhishek.das@ucl.ac.uk; deena.gibbons@kcl.ac.uk

immune perturbations from those driven by rapid immune developmental changes across gestational age (GA)[11–14]. Ultimately, a better understanding of these pathways will not only drive innovation into novel immunotherapies for sepsis in preterm babies, but additionally, may identify immune markers that can be combined with standard laboratory tests, to rule-out sepsis with greater accuracy and guide whether antibiotics need to be continued or can be stopped. This is an area of unmet clinical need, given that no current test, including the commonly assessed C-reactive protein (CRP), has sufficient accuracy to delineate sepsis when a baby first develops clinical signs[15]. This means that many babies are started unnecessarily on empirical antibiotics; antibiotic exposure in this context increases the rate of further episodes of LOS and necrotising enterocolitis (NEC)[16,17], a devastating gut inflammatory pathology of preterm babies[2]. Moreover, antibiotic exposure has been shown to impact preterm microbiome development negatively[18], and in some cases, select out resistant bacteria in the gut, propagating difficult-to-treat antimicrobial resistant infections which remain a major global health threat[19–21].

To address some of these challenges, we carried out a prospective and longitudinal study of the immune profiles of very preterm babies (<32 completed weeks GA) on the neonatal intensive care unit, the majority of whom were extremely preterm (<28 completed weeks GA). The median GA of our flow cytometry cohort was 24 weeks (range 23–29 weeks) and median birthweight was 670 g (range 460–1415 g). Through application of single-cell RNA sequencing (scRNA-seq), multi-parameter flow cytometry, and validation by plasma evaluation, our data identified distinct cellular and soluble immune traits within blood, which cumulatively provided a signature of sepsis in very preterm babies. Amongst our findings, we identified amphiregulin as an early feature of the host immune response to sepsis and a plasma analyte that correlated with infection. Together, our data provides insight into early-life sepsis-induced dysregulation and highlights a panel of immune analytes, which could be combined with current laboratory tests including CRP, to rule-out sepsis with greater accuracy.

## Results

### Clinical characteristics and disease groups

For our flow cytometry analysis, blood samples were obtained at approximately weekly intervals (a median of 9/baby) from a cohort of nineteen very preterm babies (Table 1). At each blood draw, peripheral blood mononuclear cells (PBMC) and plasma were extracted and stored for later immune-phenotyping by multi-parameter flow cytometry (Fig. 1a), and within a sub-cohort, by scRNA-seq.

To classify sepsis, two neonatologists, blinded to the immunological data, assigned each individual blood sample to one of five distinct groups dependent on contemporaneous clinical and microbiological data (detailed descriptions of each group in Fig. 1b). In total, 157 PBMC samples were analysed by flow cytometry (MCS = 26; ClinSep = 21; No Sepsis Confirmed (NSC) = 22; NEC = 6; Stable = 82; Supplementary Table 1) and 219 plasma samples were assessed for soluble analytes including cytokines (MCS = 26; ClinSep = 48; NSC = 28; NEC = 6; Stable = 111). Additionally, paired samples (PBMC or sorted CD3$^+$ T cells), obtained from five babies encountering sepsis or NEC, were analysed by scRNA-seq. Where indicated in the text and figures, MCS and ClinSep samples were combined into a composite 'Sepsis' group whilst NSC and Stable samples were combined into a composite 'No-Sepsis' group. Where CoNS (Coagulase Negative Staphylococcus) were the isolated organisms, an accompanying clinical picture which correlated with the detection of these organisms (e.g. repeated positive cultures, the simultaneous presence of a central line or culture of the organism from a central line tip) was required for a valid classification. If more than one sample was obtained during any episode of sepsis, only the first sample was included in subsequent analysis (Fig. 1c). NEC samples were not included in the flow cytometry analysis due to the limited sample size of babies with this condition. Nevertheless, two babies

with NEC were included in our scRNA-seq, and three babies in the longitudinal plasma cytokine analysis, given that this disease represents an important differential to sepsis and is a leading cause of death within this age group.

### An immune signature of sepsis

To test the hypothesis that host immune responses could differentiate Sepsis from No-Sepsis, we assessed 105 distinct immune parameters by flow cytometry in all PBMC samples (list of parameters assessed can be found in Supplementary Table 2), excluding NEC or duplicate samples from the same episode (Fig. 1c). We observed that ten immune parameters were significantly reduced in Sepsis versus No-Sepsis samples (Benjamini-Hochberg (BH) adjusted $p$-value cut-off <0.01; Fig. 1d). Features within this signature included: CD4$^+$ and CD8$^+$ T cell lymphopenia, reduced frequencies of CD3$^+$ T cells and myeloid dendritic cells (mDC), as well as diminished median HLA-DR expression within dendritic cells (DC), classical and intermediate monocytes (Fig. 1d).

We next assessed temporal changes in these signature immune traits, for each individual baby, comparing the sepsis sample to its temporally closest blood sample obtained before (median 6 days) and after (median 7 days) the event. We observed that several sepsis traits including frequencies of T cells, pDC and mDC were significantly reduced during sepsis versus the pre-sepsis timepoint (Fig. 1e). Additionally, there was a statistically significant rebound in most immune parameters approximately seven days after sepsis. Furthermore, no statistical differences between pre-sepsis and post-sepsis timepoints (approximately 14 days apart) were observed for CD4$^+$ T cell, CD8$^+$ T cell, mDC number; DC median HLA-DR and pDC and T cell frequencies, suggesting that changes observed in these parameters were mainly driven by sepsis.

To assess postnatal age as a potential confounder, we examined each of the top eight immune parameters identified above (Benjamini-Hochberg (BH) adjusted $p$-value cut-off <0.01 from Fig. 1d) longitudinally over time, restricting analysis to 'No-Sepsis' samples, to more accurately reflect normal developmental changes. Consistent with previous reports[11–14], rapid age-associated changes were demonstrable across the first weeks of life (Supplementary Fig. 1a). By grouping samples into 15-day age brackets, we observed that numbers of CD4$^+$ and CD8$^+$ T cells and frequencies of T cells and mDC significantly increased within the first month of life, whilst there was a trend towards increases in DC and classical monocyte median HLA-DR, pDC frequency and mDC counts. No significant differences were observed between other neighbouring age brackets (age brackets 2 vs 3; 3 vs 4; 4 vs 5 and 5 vs 6) for all parameters.

To ensure our dynamic immune signatures were not merely a consequence of age, we next stratified our samples into two sub-cohorts (1–30 days and >30 days) prior to comparison of Sepsis vs No-Sepsis samples (Supplementary Fig. 1b). These data still showed statistically significant reductions in CD4$^+$, CD8$^+$ T cell and mDC counts, T cell frequencies and classical monocyte median HLA-DR during sepsis in both age strata, whereas pDC and mDC frequency were significantly reduced during sepsis in the 1–30 day strata only. There was a non-statistically significant trend towards a reduction in DC median HLA-DR in Sepsis versus No-Sepsis for both age strata.

### Application of the immune signature to differentiate sepsis episodes where CRP does not rise

CRP is commonly used to support the diagnosis of sepsis, however levels have been shown to be low or undetectable (≤10 mg L$^{-1}$) in one quarter of confirmed neonatal bloodstream infections and this was more likely to occur within the extremely preterm cohort[22].

To test the hypothesis that our defined immune traits might differentiate sepsis even in settings where the CRP is low or normal, we sub-cohorted sepsis episodes into those accompanied by a CRP rise (>10 mg L$^{-1}$), and those in which CRP was normal or remained low

**Table 1 | Patient clinical characteristics**

| Baby ID | Birth GA (weeks) | Sex | Samples (n) | Day first sample | Day last sample | Microbiologically confirmed sepsis (n) | MCS Organism(s) | Late onset sepsis (LOS) vs Early onset sepsis (EOS) | Clinical Sepsis (n) | No Sepsis confirmed (n) | Stable (n) | NEC (n) | Flow cytometry | scRNA-seq | Amphiregulin FC | Plasma analysis |
|---|---|---|---|---|---|---|---|---|---|---|---|---|---|---|---|---|
| A1* | 25 | F | 10 | 1 | 48 | 3 | CoNS | LOS | 1 | 1 | 5 | - | x | | | x |
| A10* | 25 | F | 14 | 2 | 76 | 2 | E. coli | EOS | 4 | 2 | 6 | - | x | | | x |
| A11 | 29 | F | 7 | 4 | 53 | 2 | E. cloacae | LOS | - | - | 5 | - | x | x | x | x |
| A12 | 24 | M | 9 | 5 | 61 | 2 | E. cloacae | LOS | 2 | - | 5 | - | x | x | | x |
| A13 | 26 | F | 14 | 4 | 75 | 2 | CoNS | LOS | 2 | 3 | 7 | - | | x | | x |
| A14 | 24 | M | 5 | 4 | 30 | 2 | CoNS | LOS | 1 | - | 2 | - | | | x | |
| A15 | 24 | F | 13 | 6 | 76 | - | - | - | 2 | 5 | 6 | - | x | | | x |
| A16 | 26 | M | 9 | 8 | 71 | - | - | - | 1 | 1 | 7 | - | | | | |
| A17 | 23 | M | 11 | 7 | 74 | 2 | E. faecalis | LOS | 4 | 1 | 4 | - | x | | | |
| A18 | 24 | F | 10 | 5 | 69 | 2 | CoNS | LOS | 2 | 2 | 4 | - | x | | | |
| A19 | 28 | M | 8 | 8 | 56 | - | - | - | 2 | - | 6 | - | x | | | |
| A2* | 24 | F | 12 | 3 | 79 | 2 | CoNS | LOS | - | 3 | 7 | - | x | | | x |
| A20 | 24 | F | 3 | 7 | 21 | - | - | - | 3 | - | - | - | x | | | |
| A21 | 28 | M | 7 | 3 | 44 | - | - | - | 2 | 1 | 4 | - | x | | | x |
| A3* | 24 | M | 10 | 4 | 58 | 2 | CoNS | LOS | 1 | - | 7 | - | x | | | x |
| A4* | 24 | F | 3 | 2 | 8 | 1 | CoNS | LOS | - | 1 | 1 | - | x | | | x |
| A5 | 28 | F | 6 | 4 | 31 | 1 | CoNS | LOS | - | - | 4 | 1 | x | x | x | x |
| A6 | 23 | M | 5 | 8 | 35 | 1 | CoNS | LOS | - | 1 | 3 | - | x | | | |
| A7* | 26 | M | 12 | 2 | 73 | 4 | GBS/CoNS | LOS | - | 3 | 4 | 1 | x | | | x |
| A8 | 24 | M | 2 | 1 | 4 | 1 | E. coli | EOS | - | 1 | - | - | x | | | x |
| A9* | 24 | M | 14 | 2 | 84 | 2 | E. coli | EOS | 1 | 2 | 9 | - | x | | | x |
| B10 | 29 | M | 8 | 3 | 51 | - | - | - | - | - | 8 | - | | | x | x |
| B11 | 27 | F | 9 | 3 | 55 | - | - | - | - | - | 9 | - | | | x | |
| B12 | 28 | M | 7 | 6 | 48 | - | - | - | - | - | 7 | - | | | x | x |
| B13 | 29 | M | 7 | 5 | 46 | - | - | - | - | - | 7 | - | | | x | |
| B14 | 25 | M | 8 | 5 | 47 | - | - | - | - | - | 8 | - | | x | x | |
| B6 | 24 | M | 7 | 6 | 45 | - | - | - | 2 | - | 1 | 4 | x | x | x | x |
| B7 | 26 | M | 4 | 5 | 39 | 1 | S. anginosus | EOS | - | 1 | 2 | - | | | | x |
| B8 | 29 | F | 4 | 7 | 35 | - | - | - | - | - | 4 | - | | | x | |
| B9 | 30 | M | 4 | 4 | 26 | - | - | - | - | - | 4 | - | | | x | |
| C12 | 26 | M | 10 | 4 | 74 | - | - | - | 7 | 1 | 2 | - | | | | x |
| C13 | 26 | M | 9 | 3 | 43 | - | - | - | 6 | - | 3 | - | | | | x |
| C17 | 27 | M | 6 | 4 | 60 | - | - | - | 3 | 1 | 2 | - | | | | x |
| C2 | 26 | F | 9 | 4 | 87 | - | - | - | 1 | 1 | 7 | - | | | | x |
| C3 | 29 | M | 10 | 4 | 73 | 3 | CoNS | LOS | 4 | 1 | 2 | - | | | | x |
| C4 | 25 | F | 11 | 4 | 88 | - | - | - | 7 | 1 | 3 | - | | | x | x |
| C5 | 25 | F | 11 | 4 | 88 | 2 | CoNS | LOS | 4 | - | 5 | - | | | | x |

*Flow Cytometry data re-analysed from Kamdar et al. (https://doi.org/10.1038/s41467-020-14923-8).
GBS Group B streptococcus, E. coli Escherichia coli, E. faecalis Enterococcus faecalis, E. cloacae Enterobacter cloacae, S. anginosus Streptococcus anginosus, CoNS Coagulase negative staphylococcus, Amphiregulin FC flow cytometry (FC) validation of amphiregulin.

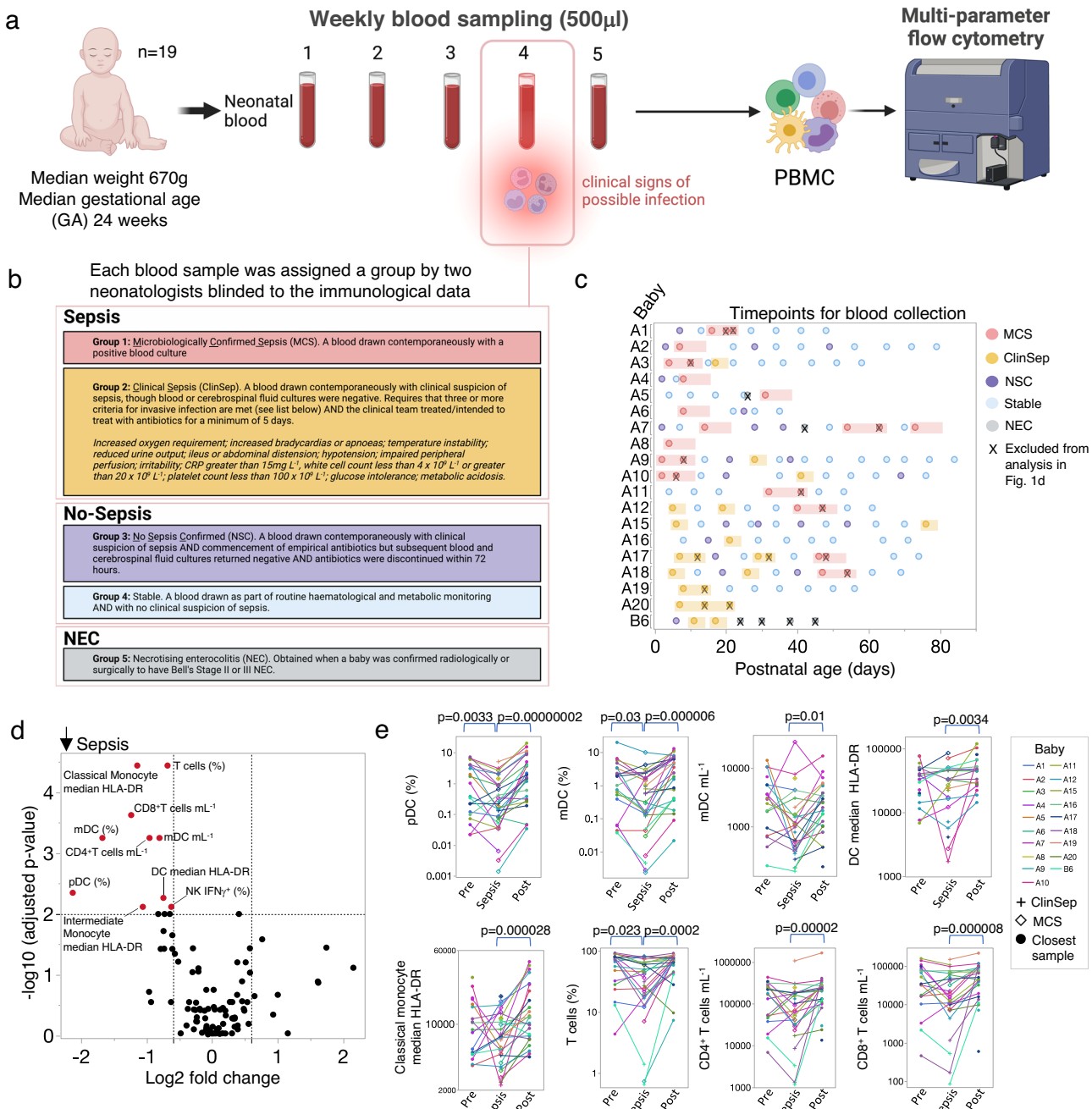

**Fig. 1 | An immune signature of sepsis in very preterm babies. a** Overview of study design. **b** Each individual blood sample from a baby was assigned to one of five groups, dependent on contemporaneous clinical and microbiological metadata. Figure created with BioRender.com. **c** Timepoints for blood collection from each baby. Repeated samples from the same sepsis episode (marked with an X), or those collected during NEC (grey circles) without positive microbiological tests were excluded from analysis in Fig. 1d. **d** Volcano plot of 105 immune parameters comparing 'Sepsis' (n = 32 samples from 19 babies) versus 'No-Sepsis' (n = 104 samples from 17 babies). Statistical analysis was performed using the two-tailed Mann−Whitney test. Statistically significant parameters are marked in red (Log2fold change >0.6 or <-0.6 and Benjamini−Hochberg (BH) corrected

*p*-value < 0.01). **e** Temporal changes in select immune parameters from (**d**), including frequencies of pDC, mDC and T cells; DC and classical monocyte median HLA-DR and numbers of CD4[+] T cells, CD8[+] T cells and mDC. Lines connect sepsis samples to those obtained approximately one week before (Pre) or one week after (Post) sepsis, from the same baby. Line colours denote which baby the sepsis episode belongs to; babies may have had more than one episode of sepsis (n = 32 episodes of sepsis from 19 babies). Two-sided Wilcoxon matched-pairs signed-rank test (Pre vs Sepsis and Sepsis vs Post) was used for (**e**). N may vary slightly between graphs due to experimental dropouts or data filtering (see "Methods"). Source data are provided as a Source Data file.

(≤10 mg L[−1]; Supplementary Fig. 2a). We observed six parameters within our core signature which demonstrated rapid temporal changes between sepsis and post-sepsis, in both 'No CRP rise' and 'CRP > 10' cohorts. These included classical monocyte median HLA-DR expression, frequencies of T cells, pDC and mDC and absolute numbers of

CD4[+] T cells and CD8[+] T cells (Supplementary Fig. 2b). These data imply that certain immune traits, either alone or together, demonstrate utility in discriminating sepsis at an early stage in preterm babies, when conventional markers such as CRP may be non-discriminatory.

## Understanding immune responses to MCS and NEC by scRNA-seq

Infections and NEC are amongst the leading causes of death in extremely preterm babies[23]. To understand whether peripheral blood immune dysregulation may shed light on the pathology of these diseases, we next applied scRNA-seq to scrutinize gene expression changes within paired PBMC samples representing timepoints across an MCS/NEC episode.

We identified four babies in whom frozen PBMC had been opportunistically collected at onset (Fig. 2a) or close to onset of a clinical event (NEC, $n = 2$, MCS $n = 2$). In each case, PBMC were thawed and analysed by scRNA-seq against the temporally closest blood draw.

In total, 14,729 cells from eight samples were analysed. Uniform Manifold Approximation and Projection (UMAP) dimensionality reduction identified ten clusters representing major immune cell populations (Fig. 2b), which were defined by expression of canonical marker genes (Fig. 2c).

## Amphiregulin is an overarching feature of the immune response to MCS and NEC

We first asked whether global changes in gene expression (i.e. within total PBMC) were detectable between MCS or NEC versus their temporally closest paired samples. This revealed *AREG*, a gene which encodes the tissue-protective growth factor amphiregulin[24], to be

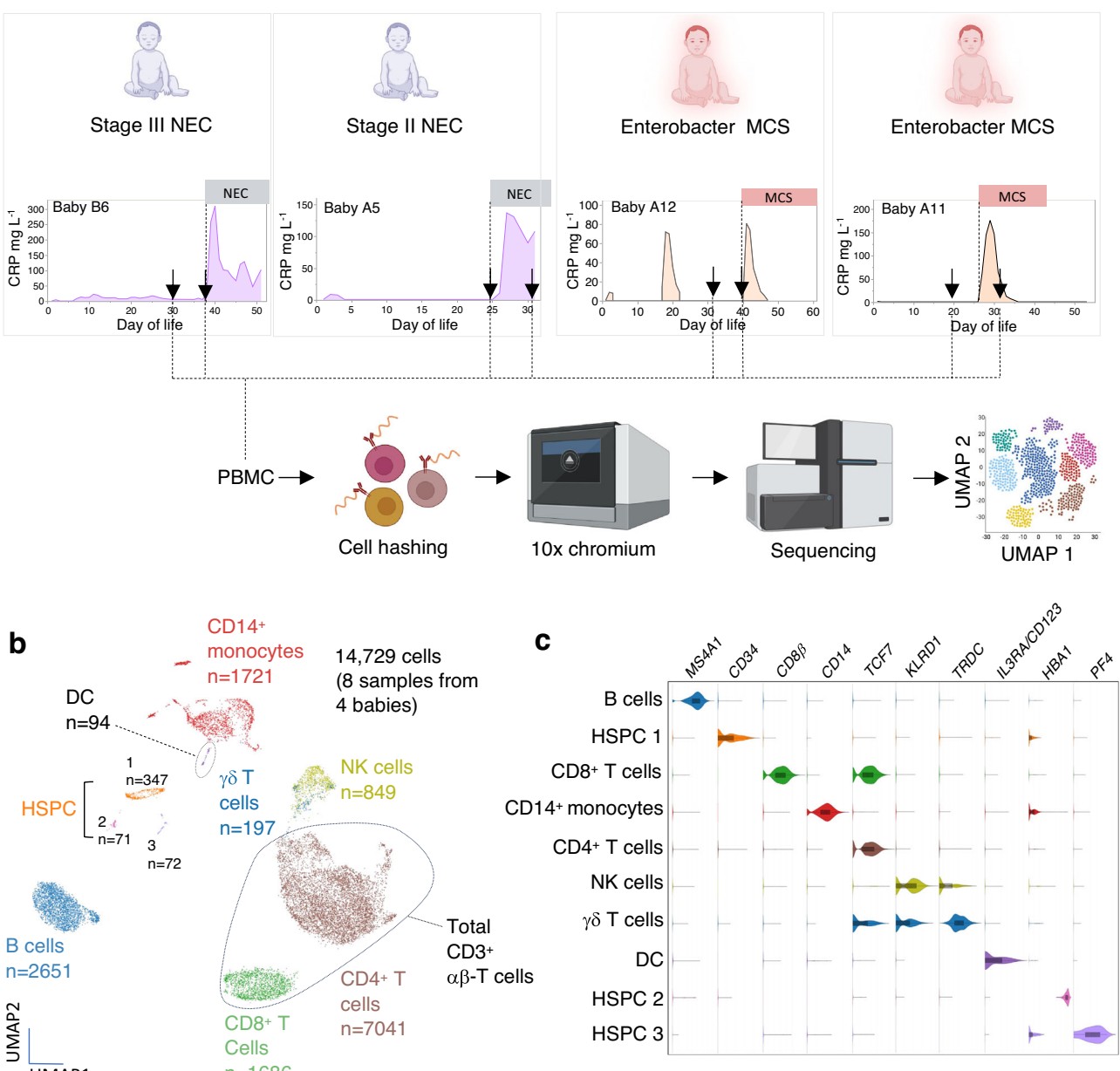

**Fig. 2 | Longitudinal scRNA-seq of PBMC in babies with MCS or NEC.**
**a** Experimental design. Paired PBMC samples were obtained from two babies with NEC and two babies with MCS. Graphs show temporal changes in CRP over time and black arrows depict when PBMC were obtained for scRNA-seq from each baby in relation to the onset of the clinical event (dotted line). Figure created with BioRender.com. **b** UMAP projection of 14,729 cells (8 samples from 4 babies). **c** Violin plots show expression of canonical gene markers (columns) for each cell cluster depicted in (**b**) (rows). Box plots; central line denotes the median, upper and lower lines of box represent the 75th and 25th percentiles respectively and whiskers show the range (minimum to maximum value). The same colours are used for corresponding cell clusters in (**b**) and violin plots in (**c**). Source data are provided as a Source Data file.

consistently within the top 50 differentially expressed genes (DEGs) in all three babies in whom we had obtained a sample at the onset of the clinical event (Fig. 3a). Conspicuously, *AREG* was not upregulated in the baby in whom sepsis was captured three days after the blood culture was confirmed positive, suggestive that the kinetics of DEGs may vary dependent on stage of disease (Fig. 3b).

We next assessed all DEGs (Benjamini–Hochberg adjusted *p*-value < 0.05, Log2 fold change >0.95 or <−0.95) that were shared or divergent between distinct NEC and MCS responses, now only focussing on the three babies for whom we had obtained samples at onset of disease. NEC stages II and III shared the highest number of

common DEGs (*n* = 143), whilst 29 DEGs were common to all three diseases (Fig. 3c). Only *AREG*, *SOCS2* (*Suppressor of cytokine signalling 2*), a negative regulator of cytokine signalling and *FKBP5* (*FK506 binding protein 5*)[25], involved in regulating intracellular glucocorticoid signalling and stress responses[26], were upregulated in all three disease settings.

To contextualise the biological pathways underpinning each disease response, we next undertook gene set enrichment analysis (generated with the web-based tool Enrichr-KG[27]), which identified apoptosis, cell death and cytokine signalling amongst six of the top ten pathways associated with Enterobacter MCS (Fig. 3d). In contrast, pathways associated with stage II NEC were almost exclusively DNA

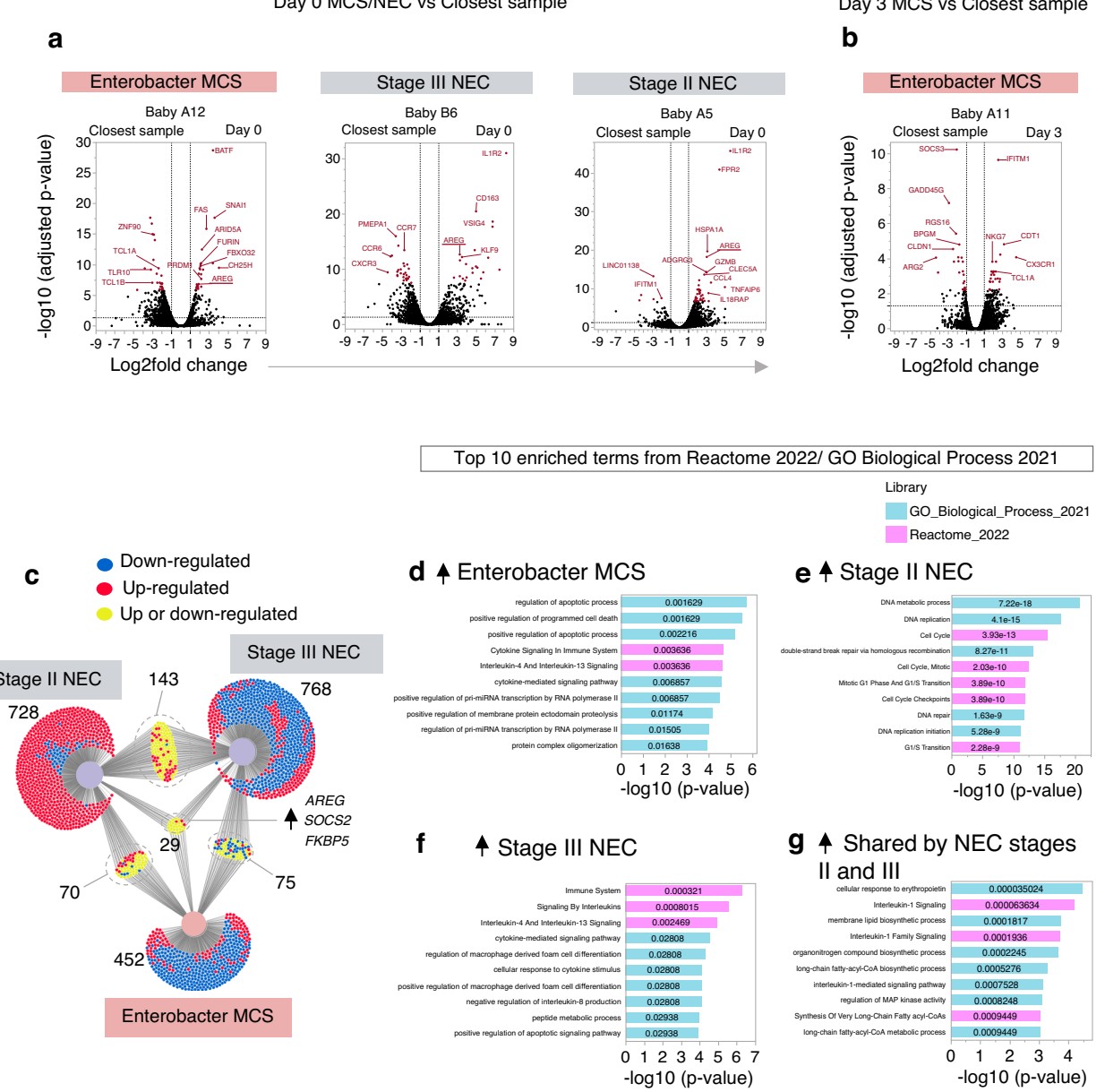

**Fig. 3 | *AREG* is upregulated during MCS and NEC.** Volcano plots show differential gene expression within PBMC obtained **a** at onset (day 0) or **b** within 3 days of onset of MCS/NEC against the temporally closest blood sample from the same baby (see Fig. 2a for timing of sampling). Differential gene expression was determined with the differential expression algorithm in Loupe Browser (10X genomics, v5.1.0), which is based on the negative binomial exact test used in the sSeq method[66]. The top 50 DEGs (by Benjamini-Hochberg (BH) adjusted *p*-values) are shown in maroon. **c**, Venn diagram (created in DiVenn 1.2) compares all DEGs (BH adjusted *p*-values < 0.05, Log2fold change >0.95 or <-0.95) between the three responses shown in (**a**). The number of DEGs in each cluster are annotated. **d–g**, Pathway analysis of all upregulated genes (BH adjusted *p*-values < 0.05) taken from distinct clusters of the Venn diagram in (**c**), as described. Bar graphs depict the top ten enriched terms from the Reactome 2022 gene set library (pink bars) or GO Biological Processes 2021 library (turquoise bars). *P*-values (Fisher exact test), adjusted for multiple testing using the Benjamini–Hochberg method, are annotated on each bar. Haemaglobin genes (*HBB, HBA1, HBA2, HBG1, HBG2, HBD, HBE1, HBM, HBQ1, HBZ*) were filtered prior to making all graphs. Source data are provided as a Source Data file.

repair or cellular proliferation (Fig. 3e), whilst 5/10 pathways relating to stage III NEC associated with interleukin signalling or cytokine responses (Fig. 3f). More DEGs were shared between NEC stages II and III, than between either NEC episode and Enterobacter MCS. Furthermore, gene set enrichment analysis of upregulated DEGs common to both stage II and stage III NEC identified pathways associated with lipid biosynthesis (4 of the top 10 pathways) and IL-1 signalling (3 of the top 10 pathways; Fig. 3g).

### NK cells and haematopoeitic stem and progenitor cells (HSPC) rapidly upregulate *AREG* during the early inflammatory response

Having identified *AREG* as an overarching DEG within PBMC at onset of sepsis or NEC, we next sought to understand its cellular source.

Highest log normalised *AREG* expression was observed in HSPC 1, DC and NK cell clusters (Fig. 4a). In contrast, *AREG* expression was barely detectable within B cells, monocytes, CD4⁺ T cell, CD8⁺ T cell and γδ T cell clusters, or other HSPC clusters (Fig. 4b). These observations are consistent with public gene datasets, albeit derived mainly from adults, which show highest *AREG* expression within haematopoietic progenitor cells and NK cells, as well as granulocytes, though we did not examine this latter population as few reside within PBMCs (meta-signature.stanford.edu; Fig. 4c)[28]. Analysis of the NK cell cluster demonstrated that *AREG* was amongst the top three DEGs (BH adjusted *p*-value < 0.05; Fig. 4d), and additionally, was the only NK DEG common to all three responses (Fig. 4e). These data point to NK cell production of *AREG* as a principal feature of early preterm inflammation.

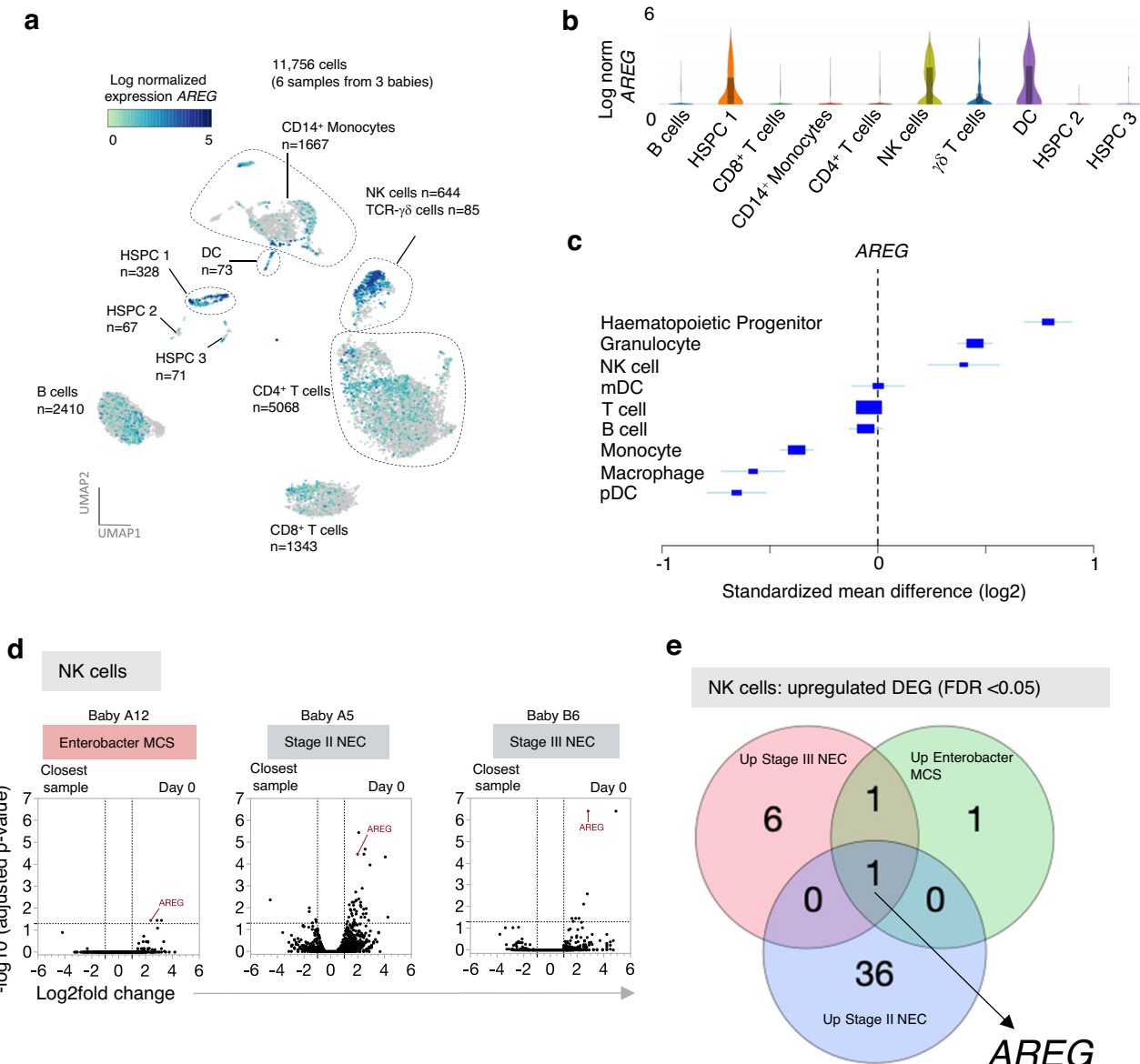

**Fig. 4 | *AREG* is highly expressed in NK cells and HSPC. a** UMAP of *AREG* expression normalized by UMI count (11,756 cells from *n* = 6 samples from babies A5, A12 and B6). **b,** Violin plots depict log normalized *AREG* expression in cell clusters from (**a**). Box plots; central line denotes the median, upper and lower lines of box represent the 75th and 25th percentiles respectively and whiskers show the range (minimum to maximum value). **c** Forest plot summarizes cell-specific expression of *AREG* from meta-analysis of public gene datasets (metasignature.standford.edu). Boxes on each row are centred at the mean effect size and are scaled to standard error. Whiskers extend to the 95% confidence interval. **d** Volcano plots show NK cell DEGs (determined with the differential expression algorithm in Loupe Browser (10X genomics, v5.1.0), which is based on the negative binomial exact test used in the sSeq method[66]) for three distinct clinical events; *AREG* is highlighted on each plot. **e** Venn diagram showing overlapping upregulated NK cell DEGs (Benjamini–Hochberg adjusted *p*-value < 0.05) between the three clinical episodes shown in (**d**). Source data are provided as a Source Data file.

## Differential gene expression within CD3$^+$ αβ T cells

Considering our immune signature of sepsis consisted of several T cell parameters, we next assessed DEGs within CD3$^+$ αβ T cells, now including a fourth baby from our cohort with CoNS MCS (associated with central line infection), for whom only sorted CD3$^+$ T cells were analysed by scRNA-seq. For this baby, a sample was obtained at onset of MCS, contemporaneous with the positive blood culture, whilst the temporally closest sample was obtained one week before MCS (Baby A13; Supplementary Fig. 3).

Interestingly, few DEGs in T cells were shared between all four episodes: Enterobacter MCS, CoNS MCS, stage II NEC and stage III NEC. The pioneer transcription factor *Basic Leucine Zipper ATF-Like Transcription Factor* (*BATF*), which is known to regulate differentiation of CD4$^+$ Th17 cells[29] and effector differentiation of CD8$^+$ T cells[30], *Fas Cell Surface Death Receptor* (*FAS*) and *AREG* were amongst the most significantly upregulated DEGs for Enterobacter MCS (Supplementary Fig. 4a), whereas the top DEGs associated with CoNS MCS, included cytokine genes (*Heparin Binding EGF Like Growth Factor* (*HBEGF*)), *Oncostatin M* (*OSM*) and *Cytokine-inducible SRC homology 2 (SH2) domain protein* (*CISH*), a negative regulator of cytokine signalling (Supplementary Fig. 4b). Upregulation of *CISH* was also the most significant DEG associated with onset of stage II NEC (Supplementary Fig. 4c), whilst *SOCS2*, another member of the same family of cytokine negative regulators, was upregulated at the onset of stage III NEC (Supplementary Fig. 4d). Though no one gene was differentially expressed across all four responses, *BATF* was significantly upregulated during CoNS MCS, Enterobacter MCS and stage II NEC (Supplementary Fig. 4e).

Gene set enrichment analysis of upregulated DEGs (BH adjusted p-value < 0.05; Log2 fold change >0.95) demonstrated that five out of ten of the top pathways during Enterobacter MCS were associated with type 1 or type 2 interferon signalling (Supplementary Fig. 4f), whereas this was not evident for CoNS MCS (Supplementary Fig. 4g). Pathways associated with stage II NEC included IL-7 signalling (2 of top 10 pathways) and type 1 or 2 interferon signalling (3 out of top 10 pathways) (Supplementary Fig. 4h), whilst NEC stage III included pathways associated with cellular responses to glucocorticoids or corticosteroids (Supplementary Fig. 4i).

## Differential gene expression within B cells

Akin to our scRNA-seq analysis of CD3$^+$ αβ-T cells, we observed that *BATF* and *AREG* were among the top differentially expressed genes within the B cell cluster during Enterobacter MCS versus its temporally closest pre-sepsis timepoint (Supplementary Fig. 5a). Other upregulated genes included the transcription factor *PR domain zinc finger protein 1* (*PRDM1*), a master regulator of terminal B-cell differentiation towards plasma-cell fate and negative regulators of cytokine signalling, *SOCS2* and *SOCS3*[25,31]. This latter observation was consistent with pathway analysis of upregulated DEGs (BH adjusted *p*-value < 0.05; Log2 fold change >0.95) which demonstrated that 8 of the top 10 pathways during Enterobacter MCS associated with cytokine signalling (Supplementary Fig. 5d).

Upregulation of *AREG* was also observed in B cells during stage III, but not stage II NEC (versus their temporally closest samples; Supplementary Fig. 5b, c). Thus, differential expression of *AREG* was evident in both Enterobacter sepsis and Stage III NEC, despite lower overall baseline expression of *AREG* in B cells compared to other cell clusters (Fig. 4b). Conspicuously, *Zinc Finger And BTB Domain Containing 16* (*ZBTB16*), the gene which encodes the transcription factor PLZF (promyelocytic leukemia zinc finger) and has been described as a suppressor of B cell proliferation[32], was amongst the top 50 upregulated genes in both cases of NEC, but not Enterobacter sepsis (Supplementary Fig. 5a–c). Pathway analysis of significantly upregulated genes demonstrated 3 of the top 10 pathways associated with TLR-4 signalling in NEC stage III (where there was clinical evidence of gut

perforation requiring surgery; Supplementary Fig. 5e) but not NEC stage II which was managed medically (Supplementary Fig. 5f).

## Preterm leukocytes rapidly induce amphiregulin

To test whether our gene expression results could be validated by flow cytometry, we assessed amphiregulin production in mitogen activated PBMC obtained from preterm babies across a range of postnatal ages. Consistent with our scRNA-seq data, HSPC (CD34$^+$ cells) and NK cells, sampled between zero and two months after birth demonstrated robust capacity to produce amphiregulin (Fig. 5a, b). Despite lower baseline *AREG* expression in most other cell clusters within our scRNA-seq analysis, we observed abundant amphiregulin induction, at the protein level, following stimulation within monocytes (mean 21.9%; Fig. 5c); B cells (mean 11%; Fig. 5d); CD4$^+$ T cells (mean 38%; Fig. 5e), CD8$^+$ T cells (mean 39%; Fig. 5f) and γδ T cells (mean 34.4%; Fig. 5g). There was no significant change in the ability of these different cell types to produce amphiregulin over the first 50 days of life (Fig. 5a–g) again suggesting the dynamic expression of amphiregulin we have noted in sepsis is not related to age.

We next assessed whether frequencies of amphiregulin$^+$ subsets might differ between preterm, cord blood and adult PBMC samples (Supplementary Fig. 6). Generally, the percentage of amphiregulin$^+$ leukocytes (NK cells, monocytes, CD8$^+$ T cells and γδ T cells) was not significantly lower in preterm compared to adult although amphiregulin$^+$ B cells and CD4$^+$ T cells were significantly lower in preterm versus either cord or adult leukocytes. These data therefore identify amphiregulin as a robust and ubiquitous function within cellular subsets of PBMC in preterm babies, with adult-equivalent levels of expression within several cell types.

Given the capacity of these different leukocyte populations to readily produce robust quantities of amphiregulin within a four-hour in vitro stimulation window, we next hypothesized that amphiregulin might be elevated in the circulation during sepsis. To first establish baseline levels in non-infected infants, we measured plasma amphiregulin in 9 babies ($n = 23$ samples) who did not encounter sepsis or NEC during their admission. This demonstrated low but detectable circulating levels (median 19 pg mL$^{-1}$, interquartile range 14–27.9 pg mL$^{-1}$) which remained consistently low over time in all the babies (n = 3/9) in whom longitudinal samples were available (Fig. 6a).

Subsequently, to assess whether plasma amphiregulin was raised in the context of sepsis, we examined amphiregulin levels over time in a cohort of 23 preterm babies who encountered one or more episodes of MCS or ClinSep, including 66 samples from 7 babies recruited from a different hospital cohort (Table 1 and Fig. 6b). Consistent with our scRNA-seq analysis, we observed precipitous increases in amphiregulin contemporaneous with sepsis, but also NEC (Fig. 6c). Notably, amphiregulin and CRP were often raised together (Fig. 6c examples: Baby C12 ClinSep; Baby A9 MCS; Baby A10 MCS; Baby A7 second MCS episode), though interestingly, we observed occasions when only one of CRP or amphiregulin was raised during sepsis (Fig. 6c examples of high amphiregulin and normal CRP: Baby A1 MCS; Baby A3 MCS or normal amphiregulin and raised CRP: Baby C3 MCS, Baby C5 MCS). This suggested that measuring both analytes together might increase sensitivity of diagnostic tests which seek to 'rule-out' sepsis.

To put our plasma amphiregulin findings in context, we next compared amphiregulin to other soluble mediators, associated in the literature with LOS in very preterm babies, namely IL-6, CXCL8, IL-10, IFN-γ and TNF-α[33–37]. Amphiregulin was the only analyte which could distinguish Sepsis samples with low or normal CRP from No-Sepsis samples (Fig. 6d), whereas amphiregulin, IL-6 and IL-10 could all distinguish No-Sepsis samples from Sepsis samples where CRP was elevated. No significant differences in levels of CXCL8, TNF-α or IFN-γ were evident between the groups of samples tested. Furthermore, sepsis-induced perturbations of amphiregulin, as well as some of the previous cellular traits we had measured, were preserved after

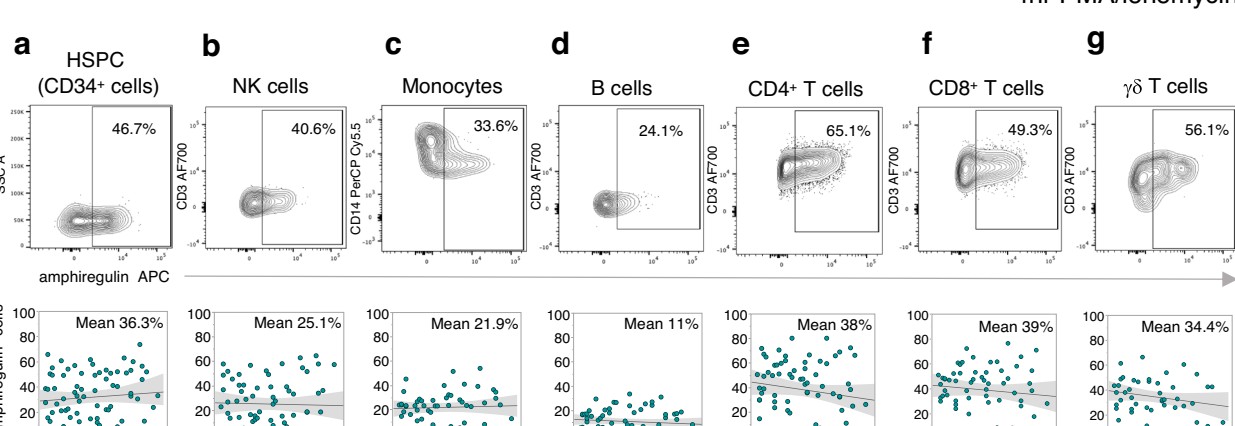

**Fig. 5 | Amphiregulin is rapidly inducible from neonatal leukocytes.** Frequency of amphiregulin⁺ **a** HSPC. **b** NK cells. **c** Monocytes. **d** B cells. **e** CD4⁺ T cells. **f** CD8⁺ T cells and **g** γδ-T cells after mitogen activation of PBMC in samples obtained across a range of postnatal ages (*n* = 72 samples from 11 babies). Top panels (**a–g**) show a representative flow cytometry plot of amphiregulin staining for each cell subset. Bottom panels (**a–g**) show a linear regression line with shaded 95% confidence intervals. Source data are provided as a Source Data file.

adjustment for known confounders, including sex, postnatal age, gestational age, birthweight Z score, and where data were available, respiratory support and antenatal steroids (Supplementary Fig. 7).

Given that amphiregulin was dynamically elevated in some cases of sepsis without a CRP rise and vice versa, we hypothesized that development of a test for sepsis that combined both parameters would increase test accuracy to 'rule-out' sepsis. We applied the commonly used cut-off of >10 mg/L for CRP[8], whilst the cut-off for amphiregulin was determined by analysis of the Receiver Operator Characteristic (ROC) curve and a threshold of 38.7 pg/ml was set, based on the maximal Youden index (sensitivity + specificity −1; Supplementary Fig. 8a)[33]. A test was designated negative if neither amphiregulin nor CRP was raised. We restricted analysis to all samples obtained within 48 h of a blood culture being taken, and antibiotics initiated, for suspected sepsis (NSC/ClinSep/MCS), as this is the window period in which the National Institute for Health and Care Excellence guidelines on antibiotic prescribing during neonatal sepsis advocate review of antibiotics for suspected infection[38].

Despite a high prior probability of infection in extremely preterm babies[1], a test based on amphiregulin and CRP together demonstrated higher sensitivity (0.97) and negative predictive value (NPV; 0.94; Fig. 6e), compared to tests evaluating CRP (Sensitivity 0.55, NPV 0.65) or amphiregulin alone (Sensitivity 0.88, NPV 0.79; Supplementary Fig. 8b), indicating a potentially greater ability of the combined test to rule-out sepsis. Given that prevalence declines with advancing birth gestation, the negative predictive value of this combined test is likely to be even higher in babies born >28 weeks or ≥ 30 weeks GA respectively, in whom the prevalence of late-onset suspected or confirmed infection has been estimated at 22% and 8% (NPV of combined amphiregulin/CRP test if prior probability of infection is 0.22 = 0.99; Supplementary Fig. 8c)[1].

## Discussion

Our study incorporated multi-parameter flow cytometry, single-cell transcriptomics and targeted plasma analysis to resolve peripheral blood immune dysregulation that may underpin the early pathophysiological response to sepsis in a cohort of predominantly extremely preterm babies. We identified core cellular and plasma immune traits that dynamically correlated with sepsis episodes and moreover, could differentiate babies with sepsis that presented with low CRP.

Our cellular signature includes lymphopenia and diminished DC and monocyte HLA-DR expression, recapitulating cardinal traits of sepsis-induced immunoparalysis, linked to adverse outcomes including mortality, in paediatric and adult cohorts[34,39]. Though data in preterm populations are sparse, our observations are in alignment with one study of very preterm babies which showed diminished numbers of non-cytotoxic T cells, amongst other subsets, after LOS[40]. We now extend these findings to other cellular subsets including mDC and pDC and moreover, demonstrate rapid kinetic perturbations within core traits, which recover to pre-sepsis levels in most cases within seven days.

Our scRNA-seq analysis of total PBMC identified a cluster of upregulated genes, shared between sepsis and NEC responses, including *AREG*, *SOCS2* and *FKBP5*. Universal upregulation of *SOCS2* likely reflects tight regulation of cytokine receptor signalling which was revealed from our gene set enrichment analysis to be amongst the top pathways associated with Enterobacter MCS and Stage III NEC and is consistent with another report of *SOCS* gene upregulation in peripheral blood of very preterm babies with LOS[41]. *FKBP5* is known to regulate glucocorticoid pathways in response to stress[26], and is upregulated in adult myeloid cells and neutrophils after incubation with plasma from septic patients[42]. Furthermore, *FKBP5* has been shown to be part of a blood transcriptomic signature of NEC, associated with severe lesions, in a preterm piglet model[43]. Indeed, though the immunopathology of NEC is distinct from sepsis, disruption of gut epithelium is frequently complicated by endotoxaemia with or without translocation of bacteria into the bloodstream, providing a potential explanation for overlapping pathways between these two disorders[44–46].

We additionally analysed specific DEGs within separate clusters of T and B cells, which identified *BATF* and *AREG* to be upregulated. This is consistent with literature which have previously identified *BATF* as part of a transcriptomic signature of neonatal sepsis, albeit from whole blood, and have incorporated this gene into sepsis scores to predict early-life sepsis[47].

To the best of our knowledge, our data provides the first description that amphiregulin is a major component of the early host response to bacterial sepsis in humans of any age, though recent proteomic analysis has implicated amphiregulin as a differentiating factor between adult critical versus non-critical Covid-19 infection[48]. Although previous studies have scrutinised whole blood gene transcription profiles in very preterm babies with LOS[41,49–51], our identification of *AREG* might partly be attributable to the strength of single-cell analysis. In our scRNA-seq data, *AREG* upregulation was mainly

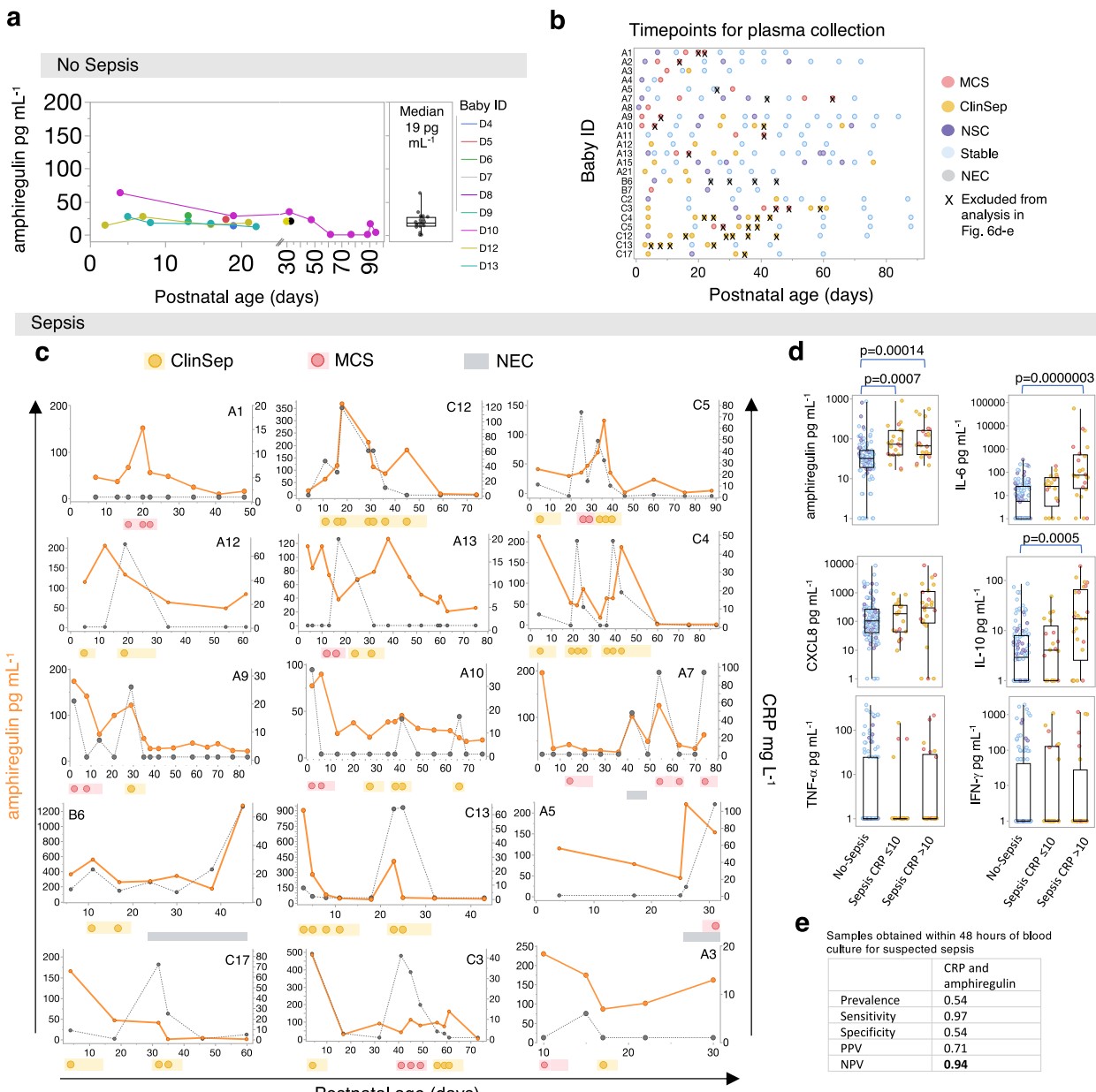

**Fig. 6 | Perturbations in plasma amphiregulin during sepsis. a** Plasma amphiregulin levels in 9 babies (*n* = 23 samples) who did not encounter sepsis or NEC during their admission. Right: Box plots; central line denotes the median, upper and lower lines of box represent the 75th and 25th percentiles respectively and whiskers show the range (minimum to maximum value). **b** Timepoints for plasma collection from each baby who encountered sepsis. Repeated samples from the same sepsis episode (marked with an X), or those collected during NEC (grey circles) without positive microbiological tests were excluded from analysis in (**d**−**e**). **c** Longitudinal changes in amphiregulin (orange line) and CRP (grey dotted line) in babies who encountered one or more episodes of sepsis or NEC (representative plots are shown for 15/23 babies; yellow circles = ClinSep; red circles = MCS; grey bars = NEC). **d** Comparison of plasma amphiregulin, IL-6, CXCL8, TNF-α, IL-10 and IFN-γ levels between 'No-Sepsis' (n = 116 samples from 23 babies) versus 'Sepsis' with normal/low CRP (≤10 mg L⁻¹; n = 21 samples from 15 babies) versus 'Sepsis' with elevated

CRP (>10 mg L⁻¹; *n* = 25 samples from 18 babies). *P*-values were generated by a Kruskal−Wallis test with Dunn's post hoc correction. Box plots; central line denotes the median, upper and lower lines of box represent the 75th and 25th percentiles respectively and whiskers show the range (minimum to maximum value). **e** Sensitivity, specificity, positive predictive value (PPV) and negative predictive value (NPV) of a test based on CRP and amphiregulin together to diagnose/rule-out sepsis (Test negative = CRP ≤10 mg L⁻¹ AND amphiregulin ≤38.7 pg mL⁻¹ where the cutoff value was defined from the maximal Youden index. Test positive = CRP > 10 mg L⁻¹ AND/OR amphiregulin >38.7 pg mL⁻¹). Analysis restricted to samples obtained within 48 h of a suspected sepsis episode (NSC *n* = 28 from 17 babies; Sepsis (ClinSep + MCS) *n* = 33 from 20 babies). Prevalence in this setting = (ClinSep + MCS)/(ClinSep + MCS + NSC); the number of suspected sepsis samples that were subsequently classified as MCS or ClinSep. Source data are provided as a Source Data file.

limited to NK cells, dendritic cells and haematopoietic stem cells, which are under-represented in whole blood gene expression profiling, and hence may have been missed previously.

Given that amphiregulin is a known tissue-repair growth factor, and inducible from multiple cell types via diverse damage-associated molecular pattern signals including ATP (adenosine triphosphate),

PGE2 (Prostaglandin E2) and cyclic AMP (adenosine monophosphate), an obvious role during preterm sepsis would be to protect developing organs from excessive tissue damage following infection-related injury[24]. In support of this, babies with homozygous mutations in EGFR, the receptor for amphiregulin and related EGFR-family ligands, develop recurrent skin and gastrointestinal infections, demonstrating

the importance of EGFR-family ligands in maintaining tissue barrier protection[52,53]. A more direct illustration that amphiregulin is important in tissue repair after infection is evident in mouse models of influenza-associated lung inflammation with or without secondary bacterial pneumonia in whom treatment with amphiregulin restored lung function and/or increased survival[54,55]. Thus, we speculate that during preterm sepsis, early induction of amphiregulin might primarily mitigate against unchecked tissue damage, critical for preservation of developing organ integrity during early life.

Our data intriguingly demonstrate, as a proof-of-concept, that immune analytes might serve as adjuncts to standard laboratory measures including CRP, to aid clinicians to rule-out sepsis with greater accuracy. The lack of ability to diagnose infections accurately at the onset of sepsis, necessitates in most cases, initiation of empirical antibiotics until further information from microbiological cultures and /or the clinical course is gained. Although UK National Guidance advocate review of antibiotics if cultures are negative within 36–48 h of suspected infection[38], in practice, many babies subsequently not proven to have bacterial infection, continue antibiotics for several days. This is partly attributable to additional variables in the neonatal setting, including concern that blood cultures may be falsely negative due to pretreatment with antimicrobials or inadequate volumes of blood collected[56]. Unnecessary or prolonged antibiotic exposure not only increases short-term risk of further episodes of NEC or LOS[16,17], but additionally has long-term implications, the most notable of which is selection of resistant bacteria which are difficult to treat and confer high sepsis-mortality[19–21]. Framed in this context, our data provides candidate markers, including those that are easy to measure in the plasma, which require further evaluation as adjunct diagnostic tests to guide rapid antibiotic de-escalation.

Our data, and those of other groups, demonstrate a prominent influence of postnatal age on immune development in early life[11–13]. Whilst we were not able to comprehensively address all other confounders due to limitations in sample sizes, our multivariate analysis demonstrated that sepsis-induced perturbations in immune traits were preserved after correction for gestational age at birth, birthweight Z-score and sex. However, we acknowledge that (i) these data require robust validation in larger cohorts, (ii) other studies have clearly documented gestational age related changes in immunity[14], and (iii) there may be other confounders including timing of infection relative to birth[51,57], type of infection (gram positive versus gram negative infection)[58] and exposure to medications including steroids which might influence the magnitude of sepsis-induced immune perturbations[34].

We acknowledge that a further limitation of our study is that our scRNA-seq analysis compared MCS/NEC samples against the temporally closest blood drawn from the same baby. Collecting samples from our extremely premature, very low birthweight cohort presented a real-world challenge, and reliable sampling intervals around sepsis episodes was challenging, as a baby's clinical status may have precluded taking blood for research purposes. Although we recognise this as a limitation of our work, we suggest that the paucity of data on sepsis-induced dysregulation in extremely preterm cohorts likely reflects this universal challenge. Nevertheless, we were able to collect four samples for scRNA-seq on the same day that MCS or NEC were diagnosed and we focussed subsequent pathway analyses on these samples. Our scRNA-seq analysis furthermore focussed on larger cell populations including T cells and B cells which were predominant in our samples and within the UMAP. Analysis of less well represented cell subsets within PBMC, including dendritic cells, was limited due to the small volume of blood obtained from very low birthweight babies (<500 microlitres).

Finally, we recognise the challenge of implementing rapid-flow cytometry based assays into the workflow of routine hospital laboratories as a diagnostic tool, mainly due to the requirement of expertise to analyse and interpret such data in real-time. However, given that current multi-centre trials of antibiotic stewardship have now included flow cytometry based assessment of sepsis-associated immunosuppression, including T cell frequency and monocyte HLA-DR, translation of this methodology into clinical practice is likely to become increasingly accessible[59].

In summary, our data identify cellular immune traits which differentiate Sepsis from No-Sepsis samples obtained from preterm babies, including those with low or undetectable CRP responses. Secondly, we identify amphiregulin as an easily measured soluble factor associated with inflammation in preterm babies and a hitherto unidentified robust function of preterm leukocytes. These data highlight candidate immune markers, which combined with current laboratory tests, could provide an innovative approach to development of diagnostic tests that rule-out sepsis with greater accuracy. Such developments, pending validation in larger cohorts, could potentially guide rapid antibiotic de-escalation and reduce onward risk of NEC, LOS and antimicrobial resistant infections.

## Methods

### Study design, ethical approval and funding

Ethical approval was granted by the London (Chelsea) Research Ethics Committee (Ref: 15/LO/1924) and the Research and Innovation Department at Homerton Healthcare NHS Foundation Trust. Babies born before 32 weeks gestation were prospectively recruited from the neonatal intensive care unit at Homerton Healthcare NHS Foundation Trust between January 2016 and August 2018, following written informed consent from their parents, which was sought within the first 72 h after birth. Translation services were available if English was not the parent's first language. Babies with known exposure to HIV or hepatitis B virus infection were excluded. Blood samples were obtained at weekly intervals from each baby, and additionally, upon suspicion of possible sepsis or NEC. Plasma and extracted PBMC were stored. No statistical method was used to predetermine sample size and the experiments were not randomized although the clinical status was not known at the time of flow cytometric analysis. A sub-cohort of thirty preterm babies were recruited from this site. Twenty-one babies were born extremely preterm (GA less than 28 weeks) whilst nine babies were born between 28 and 32 weeks gestation. There was no predominance of either males or females; 9/19 (47%) babies included in the flow cytometry cohort were female and 11/23 (48%) of babies included in the plasma analysis were female. Clinical details for all babies can be found in Table 1. Investigators carrying out the assays were blinded to the clinical data until the end of the study, except for the scRNA-seq analysis, for which stored samples from babies known to have sepsis were unblinded to the investigators. For validation of plasma amphiregulin, additional blood samples were obtained longitudinally from a further seven babies (n = 66 samples) on the Neonatal Intensive Care Unit at Guy's and St Thomas' NHS Foundation Trust following written informed consent [Ethical approval granted by the London - Fulham Research Ethics Committee, Ref: 20/PR/0964]. For validation of amphiregulin levels in stable babies, we included 23 plasma samples from 9 babies who did not experience either sepsis or NEC during their admission.

Our analysis incorporated flow cytometry data from seven babies who were included in a parent study derived from the Homerton patient cohort, Kamdar et al. (doi: 10.1038/s41467-020-14923-8.; see Table 1)[11], representing approximately one third of the flow cytometry cohort described here. In each case, FCS files were downloaded (http://flowrepository.org/id/FR-FCM-Z2FJ), re-compensated and re-gated. Cord blood samples were purchased from the Anthony Nolan Research Tissue Biobank (Cell and Gene Therapy Services) after informed consent [Ethical approval NRES Committee East Midlands - Derby Ref 15/EM/0045]. Control adult blood samples were collected

after informed consent [Ethical approval granted by the London - Fulham Research Ethics Committee, Ref: 20/PR/0964].

Funding was granted by Barts Charity (Ref: 764/2306) for the undertaking of a study named 'Investigating Microbial Colonisation and Immune Conditioning in Preterm Babies' and funding from Action Medical Research (Ref:GN2790) for a study named 'Identifying immune correlates of infection in preterm infants'.

## Clinical definitions

Data from health records were collected prospectively for all babies in the study. Each blood sample was then grouped into one of five categories based on contemporaneous clinical metadata, laboratory tests and microbiological culture results. All categorizations were assigned jointly by two clinical neonatologists involved in the study.

1. Microbiologically Confirmed Sepsis (MCS): A blood drawn contemporaneously with a positive blood culture.
2. Clinical Sepsis (ClinSep): A blood drawn contemporaneously with clinical suspicion of sepsis, though blood or cerebrospinal fluid cultures were negative. Requires that three or more criteria for invasive infection were met (see list below) AND the clinical team treated/intended to treat with antibiotics for a minimum of five days.

   Increased oxygen requirement; increased bradycardias or apnoeas; temperature instability; reduced urine output; ileus or abdominal distension; hypotension; impaired peripheral perfusion; irritability; CRP greater than $15\,mg\,L^{-1}$, white cell count less than $4 \times 10^9\,L^{-1}$ or greater than $20 \times 10^9\,L^{-1}$; platelet count less than $100 \times 10^9\,L^{-1}$; glucose intolerance; metabolic acidosis. Criteria are taken from the ELFIN trial[1].
3. No Sepsis Confirmed: A blood drawn contemporaneously with clinical suspicion of sepsis AND commencement of empirical antibiotics but subsequent blood and cerebrospinal fluid cultures returned negative AND antibiotics were discontinued within 72 h.
4. Stable: A blood drawn as part of routine haematological and metabolic monitoring AND with no clinical suspicion of sepsis.
5. Necrotising enterocolitis (NEC): Obtained when a baby was confirmed radiologically or surgically to have Bell's Stage II or III NEC[60].

## Sample collection and storage

PBMCs were isolated from 0.5 ml of whole blood by Ficoll (GE Healthcare) separation, frozen in Cryostore (Sigma) and stored at −80 °C. Plasma was either stored from this sample or additionally, a further 0.5 ml of blood was centrifuged at $1500 \times g$ for 10 min and plasma was aliquoted and stored at −80 °C. Cord blood mononuclear cells were isolated from cord blood as above. Adult peripheral blood was obtained from five healthy donors and processed as above.

## Flow cytometry staining and acquisition

PBMC were thawed and assessed by flow cytometry on the same day. Cells were resuspended in FACS buffer (phosphate-buffered saline (PBS) plus 0.5% heat-inactivated foetal calf serum and 2 mM EDTA; Invitrogen) and divided into nine aliquots for cell staining. Seven flow cytometry panels were designed (Supplementary Table 3) to assess T cell proliferation and memory status (Panels 1–2); T cell, γδ T cell and NK cell phenotype, activation status and function (Panels 3–5), and myeloid and B cell features (Panels 6–7). For each functional panel, cells were plated in duplicate to include an unstimulated control. In samples which required both scRNA-seq and flow cytometry from less than 500 μl of whole blood, a limited functional panel was substituted for panels 4 and 5 (see Supplementary table 3). Amphiregulin was assessed by intracellular cytokine staining in a separate validation cohort of babies, which included some of those, in whom scRNA-seq was performed (Panel 8). Flow cytometry gating strategies for all panels are shown in Supplementary Figs. 9–14. Although freezing PBMC might affect myeloid and or other cell-type viability, all samples from an individual baby were batch thawed and assessed by flow cytometry on the same day to reduce any variability.

Prior to surface staining, 50 μl of Countbright Absolute counting beads (C36950, Life Technologies) were added to one aliquot from each sample to allow for cell enumeration. Cells were then washed with 100 μl Dulbecco's PBS (1x, Gibco) and resuspended in PBS containing Zombie NIR Fixable Viability dye (1:1000 dilution; Biolegend) for 15 min at room temperature in the dark, with the addition of TCR Vδ1-FITC (TS8.2; Thermo Fisher) for panels 1 and 2. All panels were then washed twice with 150 μl eBioscience FOXP3 fixation/permeabilization buffer (FPB; Invitrogen) for panels 1 and 2, or FACS buffer for panels 3 to 8, and then resuspended with FPB alone (panels 1–2) or their respective surface marker antibody cocktail in FACS buffer for 20 min at room temperature. All centrifugation steps prior to fixation were carried out at 400 g for 5 min. All steps after fixation were performed at 2,000 RPM for 1 min. After a further two subsequent washes, cells were resuspended in 100 μl of FPB with FoxP3 antibody for panels 1–2 or 1× Cell Fix (BD) for panels 3–8, for a further 30 min at room temperature. After fixation, panels 1–3, 6 and 7 which did not require intracellular cytokine staining, were washed, resuspended in FACS buffer and acquired on a four laser BD LSRFortessa X20 Flow cytometer using FACSDiva (V8; BD) software. For all other panels, an antibody cocktail to stain for intracellular proteins was added in 1× permeabilization buffer (Biolegend) for 30 min at room temperature in the dark. Cells were then washed twice in permeabilization buffer and resuspended in FACS buffer for acquisition.

## Flow cytometry data analysis

Data were analysed using FlowJo (v10.6.2; BD) Software. All gating strategies are shown in Supplementary Figs. 9–14. As a quality control step, cell populations were excluded from downstream analysis if the event count in the parent population did not meet a minimum threshold of 30 events.

## Cell stimulation

For panels 4, 5 and 8, PBMC were stimulated for 4 h at 37 °C in the presence 20 μg/ml brefeldin A (BFA; Sigma) alone (to assess spontaneous cytokine production) or BFA in addition to 10 ng/ml phorbol myristate acetate (PMA; Sigma) and 1μg/ml ionomycin in complete medium (CM: RPMI-1640 (Invitrogen), 10% Fetal Calf Serum (FCS; StemCell Technologies), 2 mM L-glutamine (Sigma), 100 U penicillin and 100 μg/ml streptomycin (Invitrogen)). We recognise that FCS itself may cause some activation but was used throughout in all cases.

## Plasma amphiregulin and cytokine analysis

Plasma amphiregulin was assessed longitudinally in serial plasma samples from 23 preterm babies with the Human Amphiregulin Quantikine ELISA Kit (R&D systems; cat no: DAR00) following manufacturer's instructions. Additionally, the following cytokines and chemokines were assessed from the same samples by multiplex bead array: IL-6, TNF-α, CXCL8, IL-10 and IFN-γ. The assay was performed using a commercially available kit (LegendPlex Human Anti-Virus Response Panel; Biolegend, cat no: 740390) as per the manufacturer's instructions with the exception that kit reagents and serum samples were diluted 2× in assay buffer. Data were analysed using the Windows LegendPlex software (v8.0, BioLegend).

## scRNA-seq cell preparation, CITE-Seq and 3' library preparation

PBMC were thawed, washed and counted as per the cell preparation protocols recommended by 10x genomics. In brief, PBMC were thawed in complete media (CM) at 37 °C and then washed twice, resuspending the cell pellet gently each time with a large bore pipette. All centrifugation steps occurred at 300 g for 5 min. After the second wash, remaining supernatant was removed and Human Trustain FcX

(Biolegend) was added in 50 µl of PBS + 1% UltraPure Bovine Serum Albumin (Thermo Scientific, cat no: AM2616) for 10 min at 4 °C. After 10 min, a cocktail of Totalseq-B preconjugated antibodies and/or hashtags were added at a concentration of 0.5 µg per test for 30 min at 4 °C (Supplementary Fig. 15). Cells were then washed twice in PBS with 1% BSA and resuspended in PBS with 1% BSA with DAPI (3µM; Biolegend, cat no: 422801) for a further five minutes before the final wash step. Live cells were sorted on a BD FACS Aria into DNA LoBind Tubes (Eppendorf) containing 20 µl of CM. After a further count with Trypan Blue Stain (Thermo Scientific), cells were loaded onto a 10x Chromium controller and 3′ gene expression, antibody derived tag (ADT) and Hash Tag Oligoneucleotide (HTO) libraries (where indicated in Supplementary Fig. 15) were prepared according to the manufacturer's instructions. Samples were sequenced with the Illumina NextSeq 500 platform, or Hiseq 2500 High Output platform dependent on the experiment.

### scRNA-seq 5′ library preparation

For baby A13, the above protocol was followed except that PBMC obtained were stained with a cocktail of Totalseq-C preconjugated antibodies; CD4 (Biolegend; cat no 300567), CD8α (Biolegend; cat no 301071) and CD27 (Biolegend; cat no 302853) or hashtags at a concentration of 1 µg per test for 30 min at 4 °C (Supplementary Fig. 15; experiment 3). Additionally, live (Zombie Aqua; Biolegend 423101) $CD3^+$ T cells were sorted on a BD FACS Aria Fusion. Sorted cells were loaded onto a 10x Genomics Chromium controller and 5′ gene expression, ADT and HTO libraries were prepared.

### scRNA-seq analysis

Single cell data were processed individually by cellranger 5.0.1 using cellranger multi mode. This mode was used to process gene expression data and antibody capture data for each sample simultaneously. Data were then further processed in R (v4.0.2), using Seurat (v4.0.1), scater (v1.18.6) and harmony (v1.0 immunogenics lab) packages[61–63]. Each sample was individually loaded into R and quality control checks were performed on the samples. The isOutlier function from scater was used to detect library size outliers, cells with low number of genes expressed and cells with a high percentage of mitochondrial reads.

For samples that were multiplexed, the HTO assay was normalized and the HTODemux function from Seurat was used to demultiplex and identify the individual babies that the cells belonged to. The samples were then filtered down to singlet cells only to remove outlier cells and the Seurat merge function was used to combine all samples together into one object.

The RNA assay from the merged object was log normalized and scaled using the default commands and parameters for Seurat (NormalizeData, FindVariableFeatures, ScaleData, RunPCA). RunHarmony was used on the RNA assay to correct the pca embeddings, with the dataset provided as the grouping variable to be removed. Further downstream steps then used the harmony embedding (RunUMAP, FindNeighbors, FindClusters).

In order to annotate the cells within our scRNA-seq dataset, we used the singleR package against the Monaco reference transcriptomic dataset[64,65]. Further manual curation was performed within Loupe Browser (v5.1.0; 10X Genomics), before being imported into R and added to the object. For differential tests, data were subset to cells of interest and the FindMarkers function was used to find only positive markers.

### Statistics and Reproducibility

JMP Pro version 17 was used for all statistical analyses. Two tailed Mann Whitney tests were used to compare unpaired samples. A P value of <0.05 was deemed significant. Where more than two samples were compared, a Kruskal-Wallis test with Dunn's post hoc correction was used. For matched samples, a two-tailed Wilcoxon matched-pairs signed rank test was used. For scRNA-seq data, differential gene expression was determined using the differential expression algorithm in Loupe Browser (v5.1.0; 10X Genomics) which is based on the negative binomial exact test used in the sSeq method[66]. Multiple comparisons between flow cytometry parameters or gene expression analyses were corrected for using the Benjamini-Hochberg method. A linear mixed-effects model fit via REML (Restricted Maximum Likelihood) was used to assess the effect of sepsis on cellular and plasma immune traits, adjusted for sex, birthweight Z-score, postnatal and gestational age, respiratory support and antenatal steroids (unless data were missing, in which case this is stated in the text). Baby ID was set as a random effect to account for repeated measures.

### Reporting summary

Further information on research design is available in the Nature Portfolio Reporting Summary linked to this article.

## Data availability

The flow cytometry data (FCS files) generated in this study have been deposited in the flowrepository database under accession code FR-FCM-Z6LE (http://flowrepository.org/id/FR-FCM-Z6LE). Our current dataset incorporates FCS files obtained from seven babies who were included in a parent study derived from the same patient cohort; Kamdar et al. (https://doi.org/10.1038/s41467-020-14923-8). For these babies, the FCS files were downloaded from http://flowrepository.org/id/FR-FCM-Z2FJ. As FCS files cannot be kept under two ID numbers on flowrepository.org, data from these 7 babies will remain under the ID: FR-FCM-Z2FJ. A list of Baby ID conversions are provided here, in order to locate the samples on flowrepository: Baby A1 = 5006; Baby A2 = 5040; Baby A3 = 5051; Baby A4 = 5060; Baby A7 = 5025; Baby A9 = 5018 and Baby A10 = 5059. Public datasets used in the present study: Meta-analysis of public gene expression microarrays to assess cell specific expression of amphiregulin was carried out using the online MetaSignature tool developed by the Khatri Lab (https://metasignature.stanford.edu/). A Forest plot was created and downloaded from the web-based tool. The scRNA-seq data generated in this study have been deposited in the NCBI Gene Expression Omnibus repository under accession code GSE236099. Source data are provided with this paper.

## Code availability

No customized code was used in the present study.

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

## Acknowledgements

Our thanks are extended to the staff of the neonatal units at Homerton Healthcare and Guy's and St Thomas' NHS Foundation Trusts, for their hard work in caring for these babies and assistance in sample and data collection, and to the families and babies who participated, for their willingness to contribute to research. We acknowledge the support of Monash Bioinformatics Platform for this work and Monash FlowCore for their assistance with cell sorting. We thank Y. Wu (KCL) for critical review of the manuscript. A.D. was supported by an Academy of Medical Sciences Starter Grant for Clinical Lecturers (SGL019\1004) and a National Institute for Health Research (NIHR) Academic Clinical Lectureship. P.F. and K.C. were both supported by a strategic research grant from Barts Charity (Grant 764/2306); D.G. and S.K. by Cancer Research UK (Grant A20730) and D.G., P.F. and G.A by Action Medical Research (Grant GN2790). S.G. was also supported by an MRC-KCL Doctoral Training Partnership in Biomedical Sciences grant (MR/N013700/1). M.S.D. is supported by an Australian Research Council (ARC) Discovery Early Career Fellowship (DE200100292) and a Rebecca L. Cooper Medical Research Foundation Project Grant (PG2020668) and an ARC Discovery Project (DP210103327).

## Author contributions

P.F. and N.G.: Patient consent and sample recruitment. A.D. and S.K.: blood processing, panel design, flow cytometry and data analysis. A.D., G.A. and M.S.D.: Single cell RNA sequencing experiments. A.D., A.B., D.D.-L. and M.S.D.: Single cell RNA sequencing data analysis. A.D. and S.G.: Plasma amphiregulin ELISA and data analysis. A.D. and G.A.: amphiregulin validation by flow cytometry and data analysis. D.G. and A.D. designed the immunological studies and evaluated the results. K.C. provided critical review of the data. A.D., P.F. and D.G. co-wrote the manuscript. P.F. and D.G. are equal contributors as last author. All authors reviewed drafts of the manuscript prior to submission.

## Competing interests

The authors declare no competing interests.
