## [Peer Review File · Nature Communications]

Identifying immune signatures of sepsis to increase diagnostic accuracy in very preterm babiesREVIEWER COMMENTS

Reviewer #1 (Remarks to the Author):

Necrotizing enterocolitis (NEC) and sepsis are leading causes of morbidity and mortality in premature infants. Limited understanding of their pathogenesis has hampered efforts to develop reliable peripheral biomarkers that can predict disease severity or guide therapies. As such improved understanding of these conditions and reliable biomarkers are urgently needed.

In this paper, Das et al investigate peripheral immune signatures related to clinical sepsis and necrotizing enterocolitis by analyzing 37 samples from infants with suspected or confirmed sepsis/NEC and 62 samples from “stable” infants. Interestingly, certain immune traits were noted during culture + sepsis and NEC that differentiated these from culture negative sepsis and “stable” infants. Importantly, some of these parameters were still detectable even in the setting of low CRP that is used by some institutions to identify patients with sepsis and could provide new means of differentiating between these conditions. Using scRNAseq of four patients, the authors identify amphiregulin (AREG) as a protein that is differentially upregulated in both NEC and culture-confirmed sepsis. They further demonstrate that neonatal leukocytes secrete (AREG) in response to stimulation and neonates with infection/inflammation have increased plasma AREG levels at the time of clinical symptoms.

The collection of these samples from micro-preemie infants and their analysis is highly commendable as this has been by far the limiting factor in understanding these conditions. However, the samples used and the way the paper is currently presented (please see below for major and minor points) do not significantly advance the field.

Major concerns:

1. Sample use: Majority of these samples have been previously published on (doi: 10.1038/s41467-020-14923-8), diminishing the novelty of the study. There are only 3 or 4 NEC cases included in the analysis (depending on which analysis is used). In only one of these cases was there NEC present without sepsis, making any conclusion about NEC beyond what is associated with sepsis highly speculative. Moreover, this does not represent the clinical scenarios observed in the NICU where only about ~25% of infants with late NEC have a positive blood culture. As such the focus of the manuscript should be on sepsis unless additional NEC samples can be analyzed.
2. Sample analysis: During the study, longitudinal samples were collected. However, the data shown is for all samples taken together? Several studies have shown that peripheral immune parameters change over time in the first months of age and as such should be analyzed in relationship to the post-natal age of the samples. This is further complicated by the fact that it looks like the LOS sepsis samples are mostly from the 80-120 d time point where the stable samples are from earlier time points, how do we know that this is not just developmental? The author's statement on line 155 that the observed immune perturbations were not associated with differences in corrected GA is not correct as Fig 1e shows a significant difference between sepsis and stable group.
3. Sample definition: The definition of culture negative sepsis and “stable” is not directly described in the manuscript and is incredibly confusing. This is fundamental to the manuscript and should be stated clearly. There is no comparison table between the infants that developed NEC/sepsis, CNS and stable infants. Were the groups comparable by GA, sex, weight, etc?

4. It would be useful to develop a predictive model based on the parameters that the authors identify differentiate sepsis from CNS/stable infants.

5. scRNAseq analysis: The comparison of samples here is highly problematic as some samples were compared to ones prior to onset of illness and some to those obtained after the onset of illness. One of the samples (e) was obtained days after the onset of illness altogether. The analysis of the data is highly limited. What about comparison to a control group, is there an increase in Areg in CNS if the CRP is high or low?

6. Areg is known to be induced in the setting of bacteremia/sepsis and produced by a large number of leukocytes, so the relevance of figure 6 is unclear. Is there a different level of Areg that is induced in preterm PBMCs vs term vs adult? The range of Areg in the CRP high and low group is almost entirely overlapping, making its utility as a biomarker rather non-specific. Is it a better predictor of CNS vs culture positive sepsis irrespective of CRP? A more indepth analysis here is warranted before the suggestion of the utility of Areg in diagnosis of sepsis as suggested in the discussion can be supported.

Minor concerns:

- The statement of 107 parameters studied is rather deceptive as very limited data is shown on immune parameters and majority of the markers used are the same in the various panels.
- The selected analysis of DEG in CD3 T cells rather than any other cell types are unclear and do not add directly to the manuscript.
- What do the numbers in the rows mean in table 1?
- Table 1: 25 weeker w/ BW 1780 g is this correct, seems unlikely or is it 780g? A couple of other weights similarly seem to be off.
- Please Comment on M/F predominance
- Figure 3: for Baby with Enterobacter sepsis, there is a spike in CRP 10 days prior to the sample listed as baseline? Any etiology for the first spike? How does this impact the analysis/interpretation of results? Did the authors consider scRNAseq on a third pair from an infant w/o CRP increase considering findings in Figure 2.
- Figure 6: What is M signify in Fig 6i

Reviewer #2 (Remarks to the Author):

The topic of the study is very important to this vulnerable group of babies. The field urgently needs specific and sensitive biomarkers to use in conjunction with CRP to minimize inappropriate use of antibiotics in sepsis and clinically suspected cases, as well as improved understanding of the immunological events that underlie sepsis and NEC.

The authors clearly have done a lot of work in a cohort that is highly challenging to study, and moreover on the topic of sepsis, which substantially adds to the challenges for multiple reasons. While I very much appreciate this effort, I have several concerns regarding the manuscript in its current form. Details are below, but my main reservations arise from the selection of groups, the criteria this selection is based upon, and therefore also the conclusions that have been drawn.

Major comments.

1. Regarding grouping of the participants in the entire paper:

a. Re cBSI: the description of this cohort lacks detail. For example, were all positive BCs included? Many (8/15) cBSIs were CONS, a notorious contaminant; were these detected on

multiple cultures to reduce the likelihood of the result being a contaminant? How were contaminants dealt with, e.g. a single CONS+ blood culture without inflammation? Were all cBSIs accompanied by inflammation, e.g. CRP rise?

b. In which cases was CS accompanied by inflammation? This is critical to know in order to enable interpretation of the data. White cell counts are notoriously non-specific; could the authors use immature to total neutrophil ratios instead?

c. The term culture-negative sepsis (CNS) is misleading, because there usually is neither sepsis nor inflammation in this setting, and antibiotics are discontinued because the treating clinicians decide in hindsight that there was no sepsis. The group the authors call CNS is a difficult group of uncertain clinical relevance, and therefore problematic as a comparator. In my opinion, this group should be disregarded entirely. Also, the term CNS is easily confused w CS in this manuscript.

d. The other groups selected for comparative analysis are also problematic. NEC is rather uncommon, whereas sepsis is common. Establishing traits of sepsis AND NEC may or may not be useful for sepsis BUT NOT NEC. A more meaningful comparison would be to analyze: cBSI either together with or separately from CS (would suggest to analyze both options) vs. NEC either together with or separately from all sepsis (all sepsis meaning cBSI+CS; again, would analyze both options) vs. stable.

e. Selection of babies for each experimental approach (i.e. FACS, sc-seq, ELISA) is confusing, without much rationale provided for why an approach was used in some, but not other babies in most cases. Some examples (but not the only ones) for this are lines 244-246 and 273-276. An exception are lines 296-298, where a clear rationale for choosing these 3 babies is given. Not having a clear understanding which data are generated from which cohort makes following the results and their interpretation quite challenging for the reader – and this situation is yet worsened by the fact that the cohorts change multiple times throughout the results section.

f. Inconsistencies with sample timing make interpretation of some of the data difficult, particularly because it is not easy to follow when which sample was taken in relation to which clinical event. I understand that the setup of the study made it difficult to achieve more scientifically relevant timing. However, the authors should please try to make it easier for the reader to understand the clinical context in many cases.

2. The abstract does not really encapsulate the findings of the paper. Could be improved by distinctly addressing background, methods, and results. There is a conclusion, but this should be revised, see my other comments in this regard.

3. Given that Fig 6. demonstrates the inducibility of AREG is strongest in T cells, the question arises whether the data analysed in Fig. 5 require some sort of normalization or consideration of cell type-proportional differences between before and after onset. For example, the data in Figs. 1&2 suggests that there are reduced T and B cell numbers, and possibly increased NK cells after onset compared to before. Therefore, the sc-RNA-seq UMAP in Fig. 5 showing AREG is potentially showing this difference, i.e. more AREG in NK cells since there are more NK cells coming from after onset, less AREG in T and B cells because there are less T and B cells coming from after onset. It would be useful for the authors to look into this and potentially revise their conclusions accordingly.

4. The findings in Figs. 1&2 suggest that there are specific cell type changes occurring in sepsis/NEC. The authors looked into this for CD3+ alpha beta T cells (Extended data 1), but do not provide a rationale for selecting this cell type, nor why the other cell types are not analyzed in a similar way.

5. In Fig. 4e, there are 26 genes that are common to cBSI and NEC. It is not apparent why only AREG was further investigated, particularly as on Line 202, the authors state that 'AREG and IL1R2, were consistently within the top 30 DEGs'.
6. Similarly, the authors do not discuss a considerable fraction of their work. For example, in Supplementary Fig 3, spontaneous and PMA/ionomycin-induced cytokine production in T/NK/TCRgd cells were assessed, but the data are not reported on. Given the changes to these cell types as demonstrated in other figures in this paper, it would be appropriate to discuss whether any functional differences were observed (or NOT observed).
7. The washing step centrifugations at 2000xg (line 472) may affect cell viability and integrity particularly in non-fixed cells. Have the authors considered and/or tested for this? Could be ok for comparisons since this approach seems to have been used throughout, but would then need to be acknowledged as a limitation.
8. Similarly, including freezing of cells in the protocol may have biased or skewed the results. In particular, data on myeloid cells that were frozen before FACS need to be interpreted with great caution due to low viability being common in this cell type in this protocol. Unless the authors have data to show that their myeloid cell viability was high, this limitation needs to be acknowledged.
9. Please explain why smaller clusters of cells are missing from the sc-seq results, e.g. dendritic cells. Also, why are DEGs specifically discussed in T cells, but not other cell types?
10. The authors correctly state that gestational age and postnatal age affect many of the parameters they measured. Performing multivariate regression analysis (in addition to the associative analyses, eg lines 149-155) will more deeply assess the confounding effects of these parameters, thus make the data stronger. In fact, the authors should include other parameters in this multivariate analysis to assess other potential confounding factors such as respiratory support, prior infections, medications etc.
11. The discussion of functional aspects of the results as well as data synthesis should be expanded, despite the fact that this is an observational study and functional conclusions can only be inferred. Without such discussion, parts of the paper, particularly the results, read like lists of mediators without much coherence.
12. The conclusion of the authors that AREG may serve as a biomarker of inflammation in preterm babies is ultimately based on data from three individuals. Although this conclusion may well be true, the authors should substantially tone down the language they use to describe it. Similarly, other conclusions are based on a small number of babies and should be phrased accordingly.
13. Moreover, for most hospitals, it is not a realistic prospect to introduce FACS analysis as a routine clinical test that can be done quickly enough to avoid antibiotic therapy in preterm babies. Hence, even if successfully validated in larger cohorts and then implemented, it is unlikely that the authors' approach would reduce antibiotic use in the setting they call CNS. This needs to be clearly stated as a limitation. Having said this, could the authors attempt to (ideally prospectively) validate their findings using differential blood counts, which are done routinely in many hospitals? Such blood counts can measure lymphopenia...

Minor comments:

1. The use of the term “corrected gestational age” in this manuscript is not consistent with widely accepted definitions. Rather, it appears to refer to what is commonly termed as “postnatal age”. Similarly, in Fig. 1b, c, and d, “cGA” is shown in an unusual countdown fashion. If properly explained, this could be ok if desired, but labeling should please be changed.
2. In multiple places references lack commas. For example, references 19 and 20 are displayed as 1920.
3. There is inconsistency between using markers (eg CD34+) and cell subsets (eg CD8+ T cells), particularly in figures.
4. Standard nomenclature between gene and protein names is not followed.
5. Line 44: please specify VLBW weight range.
6. Line 245: please spell out the abbreviation CoNS.
7. Re “Sample Collection”: how soon after collection was the blood processed and frozen?
8. Supplementary table 1 has no legend for lettering used under each antibiotic column. One can assume that the lettering is the first letter of each antibiotic, i.e. to indicate that antibiotic was used. However in row 16, G does not match with flucloxacillin nor does F match with metronidazole.
9. Supplementary Table 1: what is the CRP column indicating? Number of CRP records? Highest CRP value? Something else?
10. Supplementary table 2: A description of which cell type each marker is identifying would be useful.
11. Fig. 1: correlation method is not stated, nor indication that 95% CI is shown.
12. Fig. 5c: Bars don't line up well with cell type text.
13. Fig. 6i-k: Unclear if shaded graph is CRP or AREG
14. Supplementary Table 3: Jan-50 in dilution column
15. Were the senior physicians assigning the sepsis/NEC definitions blinded to the other data? Please state.
16. 10% FCS is not an ideal stimulation condition, as FCS can substantially activate primary cells such as PBMC. 10% FCS was used throughout all stimulation conditions, so data could be ok for comparisons, but generalizations need to be made very carefully and this limitation should please be acknowledged.
17. Referencing should be expanded to include other important work done in the field.
18. There must be an error with some of the birth weights in table 1. Baby A1 surely was not 1780g at birth at 25 weeks. The same question for A7, 1113g at 26 weeks, and B8 and B13

are also doubtful.

Reviewer #3 (Remarks to the Author):

This is an excellent paper. The study is well conducted, well written and important.

NEC and sepsis are the leading causes of mortality in babies born at extreme prematurity. Diagnostic discrimination in clinical practice can be challenging and is a major unmet need in implementation of early effective treatment. This is compounded by poor understanding of underpinning disease biology. This study uses state of the art immune profiling techniques (multi-parameter fluorescence cytometry, peripheral blood sc transcriptomics and ELISA) to study NEC and sepsis in this patient group, identifying amphiregulin as a potential plasma biomarker.

The supplementary data depicts robust flow cytometry gating strategies. The methodology is sound and are provided in enough detail for replication.

There are some potential limitations that should be discussed or clarified if already considered and addressed in the presented dataset:

- The sample size is small, more so once sub-stratified for the various experimental assays (i.e. few babies were phenotyped by more than one assay). However, it is notable how complex it is to sample babies of this prematurity in sufficient volume to then undertake multi-parameter phenotyping.

- Linked to above I think many figures amalgamate multiple samples from the same donor. Is it possible that significant findings are driven/weighted by a large number of samples being derived from a single donor in either the healthy or disease cohort? Should matched samples in disease and non-disease groups that originate from the same baby be highlighted in some figures e.g. figure 2 boxplots, and figure 6H for example. I would be interested in the authors' thoughts on this.

I have one other minor suggestion in that the abstract would be strengthened by defining extreme prematurity

REVIEWER COMMENTS

Reviewer #1 (Remarks to the Author):

Necrotizing enterocolitis (NEC) and sepsis are leading causes of morbidity and mortality in premature infants. Limited understanding of their pathogenesis has hampered efforts to develop reliable peripheral biomarkers that can predict disease severity or guide therapies. As such improved understanding of these conditions and reliable biomarkers are urgently needed.

In this paper, Das et al investigate peripheral immune signatures related to clinical sepsis and necrotizing enterocolitis by analyzing 37 samples from infants with suspected or confirmed sepsis/NEC and 62 samples from “stable” infants. Interestingly, certain immune traits were noted during culture + sepsis and NEC that differentiated these from culture negative sepsis and “stable” infants. Importantly, some of these parameters were still detectable even in the setting of low CRP that is used by some institutions to identify patients with sepsis and could provide new means of differentiating between these conditions. Using scRNAseq of four patients, the authors identify Amphiregulin (AREG) as a protein that is differentially upregulated in both NEC and culture-confirmed sepsis. They further demonstrate that neonatal leukocytes secrete (AREG) in response to stimulation and neonates with infection/inflammation have increased plasma AREG levels at the time of clinical symptoms.

The collection of these samples from micro-preemie infants and their analysis is highly commendable as this has been by far the limiting factor in understanding these conditions. However, the samples used and the way the paper is currently presented (please see below for major and minor points) do not significantly advance the field.

Major concerns:

1. Sample use: Majority of these samples have been previously published on (doi: 10.1038/s41467-020-14923-8), diminishing the novelty of the study.

We thank the reviewer for their comments which we hope they agree we have now addressed in full.

Due to the difficulty in obtaining blood samples (as highlighted by the reviewer) from extremely premature babies with sepsis, we had, in our initial manuscript, incorporated some published data from Kamdar et al. (doi: 10.1038/s41467-020-14923-8) which we fully acknowledged. However, our analyses of these data were completely different to that in the published work. Whereas Kamdar et al. grouped all longitudinally collected blood samples from the same baby into one category, our re-analysis treated each individual blood draw as a separate entity, thus contextualising each sample to contemporaneous clinical or microbiological events. Nevertheless, we agree with the reviewer that addition of new data would further improve this study. Subsequently, we now include longitudinal analysis from a further 6 new babies (which equates to 53 additional samples), all of whom encountered one or more episodes of sepsis. These additional data expand the number of samples in our flow cytometry cohort by 50%, which significantly improves the robustness of our findings. As a result, the previously published samples now represent approximately one third of our flow cytometry cohort.

There are only 3 or 4 NEC cases included in the analysis (depending on which analysis is used). In only one of these cases was there NEC present without sepsis, making any conclusion about NEC beyond what is associated with sepsis highly speculative. Moreover, this does not represent the clinical scenarios observed in the NICU where only about ~25% of infants with late NEC have a

positive blood culture. As such the focus of the manuscript should be on sepsis unless additional NEC samples can be analyzed.

We have taken this reviewer comment on board and have made a number of changes to the manuscript in response. Firstly, we have modified the title of the manuscript to focus on sepsis and not NEC. Secondly, for clarity, we have removed the few NEC samples from our flow cytometry data analysis in order to focus on sepsis-related perturbations only. Thirdly, although we do comment on NEC in our subsequent scRNA-seq and plasma amphiregulin analysis, we have no longer combined 'sepsis' and 'NEC' samples into the same composite group. Thus, we feel that this point has been fully addressed.

2. Sample analysis: During the study, longitudinal samples were collected. However, the data shown is for all samples taken together? Several studies have shown that peripheral immune parameters change over time in the first months of age and as such should be analyzed in relationship to the post-natal age of the samples. This is further complicated by the fact that it looks like the LOS sepsis samples are mostly from the 80-120 d time point where the stable samples are from earlier time points, how do we know that this is not just developmental? The author's statement on line 155 that the observed immune perturbations were not associated with differences in corrected GA is not correct as Fig 1e shows a significant difference between sepsis and stable group.

We fully agree with the reviewer's comment and apologise that we had not addressed this appropriately in the initial submission. We have now made two notable changes to our figures.

In the new Extended Data Figure 1, we now clearly depict postnatal age-associated changes for each parameter in our sepsis signature, and statistically compare samples which have been grouped into six distinct 14-day age 'brackets'. In most cases, although we noted significant age-related changes over several weeks/months, as is well documented in the literature, there was no statistically significant difference between neighbouring 14-day age brackets in by far the majority of cases. To relate these data to our proposed sepsis signatures, we now include a new main figure which depicts longitudinal changes, before during and after sepsis over an approximate 14-day window period (Fig. 2a). As we identify significant changes in this short time frame, we can conclude that these changes are sepsis driven as we did not see significant changes in an equivalent time frame due to age alone. In most cases (numbers of CD4⁺ T cells, CD8⁺ T cells and mDC; frequencies of T cells, pDC and IFN- γ ⁺ NK cells; DC median HLA-DR), there was also no statistical difference between the pre and post-sepsis timepoints, further suggesting that changes were likely driven by sepsis.

3. Sample definition: The definition of culture negative sepsis and "stable" is not directly described in the manuscript and is incredibly confusing. This is fundamental to the manuscript and should be stated clearly. There is no comparison table between the infants that developed NEC/sepsis, CNS and stable infants. Were the groups comparable by GA, sex, weight, etc?

We apologize for the lack of clarity in the sample groupings, commented on by two reviewers, which we agree is critical for understanding the manuscript findings. We have now changed the groupings and have clearly explained them. We have also simplified the nomenclature to make the abbreviations less confusing. This is now presented in Figure 1, rather than in just the methods section to make it much more accessible for the reader. We copy the table below for reference.

Sepsis

Group 1: Microbiologically Confirmed Sepsis (MCS). A blood drawn contemporaneously with a positive blood culture

Group 2: Clinical Sepsis (ClinSep). A blood drawn contemporaneously with clinical suspicion of sepsis, though blood or cerebrospinal fluid cultures were negative. Requires that three or more criteria for invasive infection are met (see list below) AND the clinical team treated/intended to treat with antibiotics for a minimum of 5 days.

Increased oxygen requirement; increased bradycardias or apnoeas; temperature instability; reduced urine output; ileus or abdominal distension; hypotension; impaired peripheral perfusion; irritability; CRP greater than 15mg L⁻¹, white cell count less than 4 x 10⁹ L⁻¹ or greater than 20 x 10⁹ L⁻¹; platelet count less than 100 x 10⁹ L⁻¹; glucose intolerance; metabolic acidosis.

No-Sepsis

Group 3: No Sepsis Confirmed (NSC). A blood drawn contemporaneously with clinical suspicion of sepsis AND commencement of empirical antibiotics but subsequent blood and cerebrospinal fluid cultures returned negative AND antibiotics were discontinued within 72 hours.

Group 4: Stable. A blood drawn as part of routine haematological and metabolic monitoring AND with no clinical suspicion of sepsis.

NEC

Group 5: Necrotising enterocolitis (NEC). Obtained when a baby was confirmed radiologically or surgically to have Bell's Stage II or III NEC.

We have shown the demographic characteristics of all babies in table 1. Unfortunately, as each sample from a baby is categorised independently, based on contemporaneous clinical data available at the time of the blood draw (the time course of sampling for each baby is depicted more clearly in a new Figure 1c), we are not able to compare demographic characteristics between samples.

4. It would be useful to develop a predictive model based on the parameters that the authors identify differentiate sepsis from CNS/stable infants.

We agree with the reviewers that a predictive model would be valuable. Below we show Receiver Operator Characteristic (ROC) curves to classify 'Sepsis' on the basis of CRP alone (area under the curve (AUC) = 0.76) versus CRP + the top 10 sepsis-induced immune traits (AUC = 0.94). This demonstrates that, in principle, there is added value to incorporating immune characteristics into predictive scores for sepsis in preterm, very low birthweight cohorts. We have included these data for the reviewers, however, have not incorporated these data into the manuscript as they remain preliminary and we could not include our novel amphiregulin finding (as different samples were used). Our aim is to assess the applicability of this model in a much larger validation cohort which is the focus of future work and therefore be able to incorporate amphiregulin into this score.

5. scRNAseq analysis: The comparison of samples here is highly problematic as some samples were compared to ones prior to onset of illness and some to those obtained after the onset of illness. One of the samples (e) was obtained days after the onset of illness altogether. The analysis of the data is highly limited. What about comparison to a control group, is there an increase in Areg in CNS if the CRP is high or low?

We thank the reviewer for this point. Our ethics allowed for samples to be obtained only at timepoints when blood was being taken for routine clinical care. However, at the point that sepsis was first suspected, or when a pre or post sepsis sample was desirable for comparison, a baby's clinical condition may have precluded an additional research blood draw. Thus, timing of research sampling was implemented to the best of our ability, and indeed in four out of five cases, we were able to obtain critical samples at the onset of disease (sepsis or NEC) for scRNA-seq analysis. In the one case in which the 'sepsis' sample was obtained three days after the onset of infection, we have clearly indicated this in the text, Figure 4b, and Supplementary Table 1. Additionally, we have discussed the differences between the signatures in this mid-sepsis sample, with those obtained at onset of an event (the lack of amphiregulin signature for example). We agree with the reviewer that this is a limitation of the study and we have now clearly stated this within our discussion (line 397). However, we re-iterate the reviewer's own comments earlier in the text that collection of samples from extremely fragile, very low birthweight premature cohorts remains a real-world challenge. This likely underpins why babies at this extreme of prematurity are poorly studied and why few studies have even attempted to longitudinally immunophenotype preterm babies across sepsis episodes.

Although we do not have remaining samples to repeat the scRNA-seq analysis, we have extended our longitudinal plasma amphiregulin analysis to a new cohort of seven babies at a different hospital, which provides robust validation of rising plasma amphiregulin, albeit at the protein level, associated with sepsis onset (and additionally, in the few cases of NEC included in the study). Thus, we can now compare plasma amphiregulin levels in high and low CRP sample groups, as per the reviewer's question. We are pleased to report that amphiregulin was elevated in sepsis in this new cohort and indeed could distinguish sepsis even in patients with low CRP where other analytes previously associated with sepsis (e.g. IL-6, CXCL8) could not (new figure 6i).

6. Areg is known to be induced in the setting of bacteremia/sepsis and produced by a large number of leukocytes, so the relevance of figure 6 is unclear. Is there a different level of Areg that is induced in preterm PBMCs vs term vs adult? The range of Areg in the CRP high and low group is almost entirely overlapping, making its utility as a biomarker rather non-specific. Is it a better predictor of CNS vs culture positive sepsis irrespective of CRP? A more in-depth analysis here is warranted before the suggestion of the utility of Areg in diagnosis of sepsis as suggested in the discussion can be supported.

To the best of our knowledge, our data provide the first demonstration that amphiregulin is raised concurrent with bacterial sepsis in humans, in either paediatric or adult cohorts. We are happy to include any published data we may have missed if the reviewer can direct us to these. Recent proteomic studies in Covid-19 have now associated critical SARS-CoV-2 infection with amphiregulin and we have now cited this article within the current manuscript (line 349).

Although amphiregulin is indeed known to be produced by many leukocyte subsets in adults, this analysis has not been performed in neonatal leukocytes, particularly within extremely premature babies as assessed here. In response to the reviewer's point, we now provide a new Extended Data Figure 3, which shows that the frequency of amphiregulin positive leukocytes reach adult-levels in blood samples obtained from preterm babies, even in the first week of life. The addition of our new infant cohort further supports the concept that amphiregulin could be a useful biomarker and these additional samples have segregated the 'Sepsis with low/normal CRP group ($\leq 10 \text{ mg L}^{-1}$)' and 'No-Sepsis' group significantly better than that which was highlighted in the initial submission.

Minor concerns:

- The statement of 107 parameters studied is rather deceptive as very limited data is shown on immune parameters and majority of the markers used are the same in the various panels.

We have shown the full list of immune markers studied in Supplementary Table 2 and have now grouped markers by immune subset for additional clarity. Although we agree that markers are shared between panels, we have excluded repeated measures such that only one of each parameter is represented. We have also clarified the number of independent parameters studied ($n=105$) in the manuscript. Additionally, we now present a new volcano plot (Figure 1d), which compares all 105 markers between 'Sepsis' and 'No-Sepsis' samples. These data now highlight significant functional differences between these groups, including reduced frequency of NK cells producing IFN- γ or TNF- α in sepsis. All 105 markers are tabulated in our source data file with sufficient metadata provided such that readers can analyse additional parameters of interest, which did not meet the log2 fold change and adjusted p-value cut-offs we assigned to select parameters within our sepsis signature (see figure 1d).

- The selected analysis of DEG in CD3 T cells rather than any other cell types are unclear and do not add directly to the manuscript.

We feel that, as T cell lymphopenia and other T cell parameters were in the immune signature that differentiated Sepsis from No-Sepsis samples, the understanding of changes in the T cell compartment in sepsis is valuable and does add another aspect to the manuscript. Furthermore, performing scRNA-seq analysis is experimentally challenging on cells extracted from such small blood volumes. As T cells make up the majority of the PBMC population we could analyse these as a separate group on Sepsis/NEC samples from an individual baby as demonstrated and shown in Extended Data Figure 2. Analysis of less well represented cell types including neutrophils and DC was technically challenging due to the small starting volume of blood (~0.5ml). We have added this as a limitation in the paper (line 408).

- What do the numbers in the rows mean in table 1?

We apologize that this wasn't clear in the table. We believe the reviewer is referring to the numbers in rows on the right-hand side of the table which refer to the number of samples from each baby assigned to one of the following groups: MCS; ClinSep; NCS; Stable or NEC.

- Table 1: 25 weeker w/ BW 1780 g is this correct, seems unlikely or is it 780g? A couple of other weights similarly seem to be off.

Apologies for the mistake in this column which is now corrected. We have checked all the other parameters listed with the original clinical data.

- Please Comment on M/F predominance

Overall, there was no predominance of sepsis in either males or females within the cohort.

Flow cytometry cohort: % Female= 47.4% (9/19 babies).

Plasma Amphiregulin/cytokines: % Female = 50% (6/12 babies).

ScRNAseq: % Female = 60% (3/5 babies).

We have now included these data in the text (line 437) and have removed the sex column from table 1.

- Figure 3: for Baby with Enterobacter sepsis, there is a spike in CRP 10 days prior to the sample listed as baseline? Any etiology for the first spike? How does this impact the analysis/interpretation of results? Did the authors consider scRNAseq on a third pair from an infant w/o CRP increase considering findings in Figure 2.

Below we attach a graph for the reviewer which shows changes in CRP and select immune parameters across ClinSep (Day 18) and Enterobacter MCS (Day 40) episodes in baby A12. As the reviewer has noted, the baby had a significant spike in CRP at day 18, which fulfilled the criteria for a ClinSep episode. The baby was treated with a five-day course of antibiotics (vancomycin and ceftazidime), after which their CRP rapidly returned to the normal range. The aetiology was suspected to be line infection, although blood cultures returned negative.

In terms of analysis, we agree with the reviewer that it is possible that this first episode of ClinSep might leave a legacy on qualitative and quantitative immune features which could impact the subsequent immune response to Enterobacter MCS. Although we cannot rule this out, for several

reasons discussed below, we believe these two episodes were distinct in their timing, aetiology and immune response.

Timing: After treatment for ClinSep, the baby's CRP returned to normal rapidly. Indeed, we collected two research samples prior to Enterobacter MCS when the CRP was completely normal, suggesting that there was a significant gap between the two episodes.

Aetiology: Although a significant CRP response $>50 \text{ mg L}^{-1}$ was raised for both episodes, their aetiologies were distinct. Whereas ClinSep was suspected to be associated with line infection, the subsequent MCS episode was associated with a significant systemic response inclusive of bradycardia, apnoea, feed intolerance, ileus and abdominal distension.

Immune profiles: Conspicuously, we noted a sharp reduction in the frequency of T cells during the first ClinSep episode, which rebounded to pre-sepsis levels within a week, and indeed, remained unperturbed for the next two weeks of 'stability' leading up to the next clinical event, Enterobacter MCS. Additionally, lymphocyte activation profiles were distinct for the two episodes. During ClinSep, there was no change in baseline activation of CD4^+ T cells, CD8^+ T cells and $\gamma\delta$ T cells, however contemporaneous with onset of Enterobacter MCS, we observed rapid activation in the same cells, preceding a CRP rise (see below graph).

Longitudinal changes in (top) CRP or (bottom) frequencies of T cells (purple), or activated (% CD69⁺) CD4⁺ T cells (green), CD8⁺T cells (orange) or $\gamma\delta$ T cells (blue) across postnatal age in baby A12. Discrete episodes of ClinSep and Enterobacter MCS are annotated. Vanc=vancomycin; Ceftaz=Ceftazidime.

Whilst a separate scRNA-seq dataset from a baby without a CRP rise would, indeed, be a great addition, we do not have samples left from such an infant on which to do this.

- Figure 6: What is M signify in Fig 6i

In the previous figure, M signified meropenem and we apologize that these abbreviations were confusing. We have now removed antibiotic annotations for clarity, however these data remain accessible in Supplementary Table 1.

Reviewer #2 (Remarks to the Author):

The topic of the study is very important to this vulnerable group of babies. The field urgently needs specific and sensitive biomarkers to use in conjunction with CRP to minimize inappropriate use of antibiotics in sepsis and clinically suspected cases, as well as improved understanding of the immunological events that underlie sepsis and NEC.

The authors clearly have done a lot of work in a cohort that is highly challenging to study, and moreover on the topic of sepsis, which substantially adds to the challenges for multiple reasons. While I very much appreciate this effort, I have several concerns regarding the manuscript in its current form. Details are below, but my main reservations arise from the selection of groups, the criteria this selection is based upon, and therefore also the conclusions that have been drawn.

Major comments.

1. Regarding grouping of the participants in the entire paper:

a. Re cBSI: the description of this cohort lacks detail. For example, were all positive BCs included? Many (8/15) cBSIs were CONS, a notorious contaminant; were these detected on multiple cultures to reduce the likelihood of the result being a contaminant? How were contaminants dealt with, e.g. a single CONS+ blood culture without inflammation? Were all cBSIs accompanied by inflammation, e.g. CRP rise?

We apologise for the lack of clarity in sample groupings, which we agree is critical to understand the paper. We have overhauled our groupings such that a) they are now straightforward to understand and b) they are standardized against other clinical groupings published in neonatal sepsis literature (e.g. ELFIN trial and nSeP trial protocols). We have provided a table of the new groupings above (in response to reviewer 1) and in the new Figure 1.

In response to the reviewer's other question:

- i) Positive blood cultures (cBSI, now renamed MCS), were included only if there was an associated antibiotic course of five days or more.
- ii) Blood cultures with Coagulase Negative Staphylococci (CoNS), a common skin commensal, were deemed significant if a) two blood cultures were positive, b) CoNS MCS was accompanied by a CRP rise or c) CoNS was additionally cultured from a central line or umbilical vein catheter tip, associated with the same episode of sepsis.
- iii) Not all cBSIs (now called MCS) were accompanied by a CRP rise. In the new Figure 2c, we examine sepsis episodes with or without CRP rise, and show that immune signatures can still differentiate MCS and ClinSep when CRP is low or normal.

b. In which cases was CS accompanied by inflammation? This is critical to know in order to enable interpretation of the data. White cell counts are notoriously non-specific; could the authors use immature to total neutrophil ratios instead?

Our ClinSep definitions were based on criteria published in the ELFIN trial (group described as 'clinically-suspected late onset infection' in their manuscript; <https://tinyurl.com/5n6mkutv>) and nSeP trial protocols (group described as 'clinical sepsis'; <https://bmjopen.bmj.com/content/11/12/e050100>).

In each case, biochemical inflammation (i.e. CRP rise) was only one of several listed criteria for ClinSep, hence the definition of ClinSep could be met without a significant inflammatory response. However, acknowledging the reviewer's point, as mentioned above, we now present a new analysis whereby MCS and ClinSep are differentiated into groups based on whether CRP was concomitantly elevated above 10 mg L^{-1} , or remained low or normal ($<10 \text{ mg L}^{-1}$). These data are shown as a new Figure 2c.

Our clinical laboratory does not routinely collect immature neutrophil counts, and therefore we are unfortunately not able to present immature to total neutrophil ratios.

c. The term culture-negative sepsis (CNS) is misleading, because there usually is neither sepsis nor inflammation in this setting, and antibiotics are discontinued because the treating clinicians decide in hindsight that there was no sepsis. The group the authors call CNS is a difficult group of uncertain clinical relevance, and therefore problematic as a comparator. In my opinion, this group should be disregarded entirely. Also, the term CNS is easily confused w CS in this manuscript.

We agree with the reviewer's assessment here and in response we have completely re-grouped our samples as shown in Figure 1. Samples previously referred to as culture-negative sepsis (CNS) have now been renamed to No Sepsis Confirmed (NSC). Additionally, for the reasons the reviewer mentions, we have now combined this group with stable samples, however they can still be differentiated by colour in the graphs (e.g. Figure 1e), so that the reader is aware that NSC samples were taken whilst a baby was on antibiotics.

d. The other groups selected for comparative analysis are also problematic. NEC is rather uncommon, whereas sepsis is common. Establishing traits of sepsis AND NEC may or may not be useful for sepsis BUT NOT NEC. A more meaningful comparison would be to analyze: cBSI either together with or separately from CS (would suggest to analyze both options) vs. NEC either together with or separately from all sepsis (all sepsis meaning cBSI+CS; again, would analyze both options) vs. stable.

As above, we have changed our groupings to combine cBSI (now MCS) and CS (now ClinSep) as suggested into a composite 'sepsis' group. Additionally, we have removed the few NEC samples from our flow cytometry data analysis to focus on sepsis-related perturbations only. Although we do comment on NEC in our subsequent scRNA-seq and plasma amphiregulin analysis, we have no longer combined 'sepsis' and 'NEC' samples into the same composite group. Thus, we feel we have fully addressed the reviewer's comment.

e. Selection of babies for each experimental approach (i.e. FACS, sc-seq, ELISA) is confusing, without much rationale provided for why an approach was used in some, but not other babies in most cases. Some examples (but not the only ones) for this are lines 244-246 and 273-276. An exception are lines 296-298, where a clear rationale for choosing these 3 babies is given. Not having a clear understanding which data are generated from which cohort makes following the results and their interpretation quite challenging for the reader – and this situation is yet worsened by the fact that the cohorts change multiple times throughout the results section.

We apologise that the sample usage was not clear and we have addressed this. Sample volume is very small from these premature babies and precludes every test being performed on every sample. Therefore, for the most part, separate cohorts were used to i) defined sepsis-induced perturbations by flow cytometry, ii) validate amphiregulin production from leukocytes by flow cytometry and iii) evaluate plasma amphiregulin/ cytokine levels. The main exception here is that for babies in whom

scRNA-seq was performed, in most cases we had sufficient paired PBMC and plasma aliquots to run flow cytometry and plasma amphiregulin/cytokine levels in parallel. Table 1 shows which babies were included for each of the assays.

f. Inconsistencies with sample timing make interpretation of some of the data difficult, particularly because it is not easy to follow when which sample was taken in relation to which clinical event. I understand that the setup of the study made it difficult to achieve more scientifically relevant timing. However, the authors should please try to make it easier for the reader to understand the clinical context in many cases.

Our ethics allowed for blood samples to be obtained only at timepoints when blood was being taken for routine clinical care. However, at the point that sepsis was first suspected, a baby's clinical condition may have precluded taking research samples and as such, due to this limitation, in some cases we were not able to obtain a sample until a few short days afterwards. Nevertheless, in most cases we were able to obtain a sample contemporaneous with symptom onset, and where that was not possible, we have clearly stated the time delay in Figure 4b and Supplementary Table 1. We have also now added a new Figure 1c, which corresponds to Supplementary Table 1, that clearly identifies when each sample was taken from each infant and the clinical state of the infant at each sampling point for the initial flow cytometry cohort. We hope this makes it easier for the reader to clearly see the sampling regime and clinical context on each baby.

2. The abstract does not really encapsulate the findings of the paper. Could be improved by distinctly addressing background, methods, and results. There is a conclusion, but this should be revised, see my other comments in this regard.

We have revised the abstract significantly, incorporating new data sets that we did not have on the original submission which have made our conclusions more robust.

3. Given that Fig 6. demonstrates the inducibility of AREG is strongest in T cells, the question arises whether the data analysed in Fig. 5 require some sort of normalization or consideration of cell type-proportional differences between before and after onset. For example, the data in Figs. 1&2 suggests that there are reduced T and B cell numbers, and possibly increased NK cells after onset compared to before. Therefore, the sc-RNA-seq UMAP in Fig. 5 showing AREG is potentially showing this difference, i.e. more AREG in NK cells since there are more NK cells coming from after onset, less AREG in T and B cells because there are less T and B cells coming from after onset. It would be useful for the authors to look into this and potentially revise their conclusions accordingly.

We agree with the reviewer that cell-type proportional changes might explain differences in AREG expression between Sepsis/NEC and their comparator samples, when PBMC are examined as a whole. However, given that we have applied single cell analysis, this limitation is abrogated if we analyse differences at an individual cell cluster level. Thus, we now show violin plots below which compare log normalized AREG expression between sepsis/NEC samples on the UMAP in Fig. 5 (n=3 samples combined) versus the temporally closest blood drawn samples (n=3 samples combined), restricted to CD3+ non- $\gamma\delta$ T cells and NK cell clusters respectively.

This shows that AREG is not strongly induced within T cells *in vivo*, at least under the specific conditions in these three babies, despite potent inducibility within the same cells upon *ex vivo* activation. We speculate that this might be a consequence of more nuanced physiological activation signals *in vivo* that preferentially activate NK cells over T cells; a feature which might not be differentiated upon activation with a maximal mitogenic stimulus.

4. The findings in Figs. 1&2 suggest that there are specific cell type changes occurring in sepsis/NEC. The authors looked into this for CD3+ alpha beta T cells (Extended data 1), but do not provide a rationale for selecting this cell type, nor why the other cell types are not analyzed in a similar way.

Performing scRNA-seq analysis on cells extracted from such small blood volumes proved to be challenging. However, as T cells make up the majority of the PBMC population, we reasoned that analysis of this cluster between paired sepsis/NEC and their temporally closest samples would be feasible, and sufficiently robust to support any conclusions generated. Secondly, T cells were a core component of our sepsis signature and thirdly, we had availability of data from a fifth baby from our cohort, in whom single cell RNA-seq analysis was performed on sorted CD3⁺ T cells only. This allowed inclusion of an additional MCS episode, in which sampling was obtained at onset (day 0) of bloodstream infection.

In addition to T cells, we also provide data in our manuscript which compare another of the larger clusters within the UMAP; NK cells. We were unfortunately not able to derive reliable data from comparison of smaller subsets including neutrophils and DC's, as their numbers in single blood draws (~0.5ml) precluded this. We have added this as a limitation in the paper (line 408).

5. In Fig. 4e, there are 26 genes that are common to cBSI and NEC. It is not apparent why only AREG was further investigated, particularly as on Line 202, the authors state that 'AREG and IL1R2, were consistently within the top 30 DEGs'.

We have taken on board point 9 of the reviewer's comments and have now included smaller cell clusters including dendritic cells and progenitor cell populations into our dataset (which were initially excluded due to lower cell numbers), which has changed some features of the scRNA-seq analysis. To that end, we now demonstrate that three core upregulated genes are common to MCS and NEC; AREG, FKBP5 and SOCS2, whilst IL1R2 remains a key upregulated gene for stages II and III NEC responses.

Our data focus on AREG because i) it was consistently within the top 40 differentially expressed genes for all three responses and perhaps more importantly, ii) no previous studies have analysed

the role of amphiregulin in sepsis in humans, particularly in the chronically understudied very preterm cohort, although there is limited data on the role of amphiregulin in tissue repair during murine viral and bacterial super-infection, which we have cited.

6. Similarly, the authors do not discuss a considerable fraction of their work. For example, in Supplementary Fig 3, spontaneous and PMA/ionomycin-induced cytokine production in T/NK/TCRgd cells were assessed, but the data are not reported on. Given the changes to these cell types as demonstrated in other figures in this paper, it would be appropriate to discuss whether any functional differences were observed (or NOT observed).

We now include a volcano plot (Figure 1d) which compares all 105 phenotypic and functional parameters analysed by flow cytometry between 'Sepsis' and 'No-Sepsis' samples. We go on to comment on the most statistically significant differences between these groups, which now mention and discuss IFN- γ expressing NK cells, for example, as these appear significantly different (Benjamini Hochberg adjusted p-value <0.01). Many other cytokine-producing cells were not different between groups and so were not discussed.

7. The washing step centrifugations at 2000xg (line 472) may affect cell viability and integrity particularly in non-fixed cells. Have the authors considered and/or tested for this? Could be ok for comparisons since this approach seems to have been used throughout, but would then need to be acknowledged as a limitation.

We apologise for the mistake in this section. The centrifugation speed should read 2000 RPM rather than G. This has now been corrected. Additionally, this speed was used only after fixation, whereas prior to fixation, cells were spun at 400g for 5 minutes (line 530). We have modified this in the methods text.

8. Similarly, including freezing of cells in the protocol may have biased or skewed the results. In particular, data on myeloid cells that were frozen before FACS need to be interpreted with great caution due to low viability being common in this cell type in this protocol. Unless the authors have data to show that their myeloid cell viability was high, this limitation needs to be acknowledged.

We did include a live dead stain in all our panels, nevertheless we agree with the reviewer that freezing might have reduced myeloid cell frequencies in all samples, and we have added this to the text as a limitation as suggested (line 517).

9. Please explain why smaller clusters of cells are missing from the sc-seq results, e.g. dendritic cells. Also, why are DEGs specifically discussed in T cells, but not other cell types?

In order to define clusters within our scRNA-seq dataset, we first used the singleR package (doi:10.1038/s41590-018-0276-y) which allowed us to perform unbiased cell type recognition against the Monaco reference transcriptomic dataset. We had not initially included these clusters in our UMAP, due to the paucity of cells (e.g. 21 neutrophils/8 basophils). However, we have now updated the UMAP in Figure 3b to include DC which appear as a distinct cluster of cells (n=94 cells), and HPSC/ progenitor cell populations 2 and 3. We attach a picture of the new UMAP below for reference.

In the main and extended figures, we show comparison of DEG in T cells, as they were the predominant cell type and NK cells because this was the cell type in which AREG was most upregulated. We did not show pairwise comparisons between samples for other cell subsets (e.g. DC) as the cell numbers within these clusters were limited, thus any conclusions drawn from these comparisons may not have been as robust.

10. The authors correctly state that gestational age and postnatal age affect many of the parameters they measured. Performing multivariate regression analysis (in addition to the associative analyses, eg lines 149-155) will more deeply assess the confounding effects of these parameters, thus make the data stronger. In fact, the authors should include other parameters in this multivariate analysis to assess other potential confounding factors such as respiratory support, prior infections, medications etc.

Whilst we agree that age has an effect on the parameters we measured, we have excluded this as the main driver behind the changes we observed in sepsis (new Extended Data Figure 1 and Figure 2a). Additionally, in response to an earlier reviewer's comment, we have also noted that the male:female ratio is approximately equal within our cohorts.

11. The discussion of functional aspects of the results as well as data synthesis should be expanded, despite the fact that this is an observational study and functional conclusions can only be inferred. Without such discussion, parts of the paper, particularly the results, read like lists of mediators without much coherence.

We thank the reviewer for this suggestion. We have now added discussion of the potential functional consequences of the lymphopenia we observe and the changes in NK and other innate cell functions which may highlight a greater usage of the innate immune system in this very early timeframe post birth.

12. The conclusion of the authors that AREG may serve as a biomarker of inflammation in preterm babies is ultimately based on data from three individuals. Although this conclusion may well be true, the authors should substantially tone down the language they use to describe it. Similarly, other conclusions are based on a small number of babies and should be phrased accordingly.

We agree that the data on plasma amphiregulin were perhaps preliminary. In order to address this, we have recruited an entirely new longitudinal sample set from a different hospital. In addition to our initial data on plasma amphiregulin, we now include a further seven babies, each of whom had a median of 10 blood samples obtained over time, including during sepsis episodes. These data confirm the trends we saw initially linking amphiregulin with sepsis, and additionally have allowed us to carry out further granular analysis between amphiregulin and sepsis timepoints. For example, we now show and discuss in the manuscript that amphiregulin, but interestingly not other common cytokines associated with sepsis (CXCL8 and IL-6) differentiate samples obtained during 'Sepsis with low CRP' from 'No-Sepsis' samples, highlighting why it might be a strong candidate to validate as a sepsis biomarker.

13. Moreover, for most hospitals, it is not a realistic prospect to introduce FACS analysis as a routine clinical test that can be done quickly enough to avoid antibiotic therapy in preterm babies. Hence, even if successfully validated in larger cohorts and then implemented, it is unlikely that the authors' approach would reduce antibiotic use in the setting they call CNS. This needs to be clearly stated as a limitation. Having said this, could the authors attempt to (ideally prospectively) validate their findings using differential blood counts, which are done routinely in many hospitals? Such blood counts can measure lymphopenia...

From our experience, most hospitals in the UK have flow cytometry capabilities although we agree the time frame required and the necessarily additional expertise to gate and analyse data, may be prohibitory and we have added this into the discussion (line 411). Nevertheless, the addition of these flow parameters to CRP, demonstrated in principle, the additive benefit of utilising such parameters in predictive scores (Nominal logistic regression to predict sepsis: CRP alone Area under curve 0.76; CRP + 10-parameter flow-based sepsis signature; AUC of 0.94; ROC curves are shown in answer to question 4 from reviewer 1). Furthermore, plasma/serum amphiregulin presents an alternative immune analyte, which is also more easily assayed, as another promising biomarker which requires validation.

Minor comments:

1. The use of the term "corrected gestational age" in this manuscript is not consistent with widely accepted definitions. Rather, it appears to refer to what is commonly termed as "postnatal age". Similarly, in Fig. 1b, c, and d, "cGA" is shown in an unusual countdown fashion. If properly explained, this could be ok if desired, but labeling should please be changed.

Apologies for the lack of clarity. We have changed all graphs to 'postnatal age'.

2. In multiple places references lack commas. For example, references 19 and 20 are displayed as 1920.

We have corrected all these errors, a problem with the referencing package used.

3. There is inconsistency between using markers (eg CD34+) and cell subsets (eg CD8+ T cells), particularly in figures.

We apologise for the inconsistency. Henceforth all cell subsets are labelled (e.g. CD34+ are now haematopoietic stem cells; HPSC).

4. Standard nomenclature between gene and protein names is not followed.

This has been corrected.

5. Line 44: please specify VLBW weight range.

We have clarified VLBW means <1500g and have amended this in the text.

6. Line 245: please spell out the abbreviation CoNS.

In the new version of the manuscript, we spell out the abbreviation CoNS when it is first mentioned (Line 106).

7. Re “Sample Collection”: how soon after collection was the blood processed and frozen?

As sepsis samples were collected at day or night by clinicians, the median time for sample collection was 7 hours (Interquartile range: Q1, 4hours; Q3, 15 hours). Any samples that were processed after 30 hours were excluded from any analysis. Whilst we are aware that some populations are affected by a delay in processing, the processing time between ‘Sepsis’ and ‘No-Sepsis’ samples compared in the cohort did not vary (see below) and so will not be driving any changes observed.

8. Supplementary table 1 has no legend for lettering used under each antibiotic column. One can assume that the lettering is the first letter of each antibiotic, i.e. to indicate that antibiotic was used. However in row 16, G does not match with flucloxacillin nor does F match with metronidazole.

We apologise this wasn't clear. We have replaced the letters with the full name of the antibiotic. Additionally, we have corrected the mistake in row 16 of the table (this row has now moved to row 78, given we have added new patients to our cohort).

9. Supplementary Table 1: what is the CRP column indicating? Number of CRP records? Highest CRP value? Something else?

The values in this column represent the CRP level on the day of research blood draw. We have changed the heading in Supplementary Table 1 to make this more clear.

10. Supplementary table 2: A description of which cell type each marker is identifying would be useful.

We have added this into the table.

11. Fig. 1: correlation method is not stated, nor indication that 95% CI is shown.

Given the change to the figures, correlation plots have been substituted with longitudinal plots showing immune changes in each parameter across different age 'brackets' (new Extended Data Figure 1).

12. Fig. 5c: Bars don't line up well with cell type text.

Thank you for pointing this out- we have realigned the bars.

13. Fig. 6i-k: Unclear if shaded graph is CRP or AREG

This figure has been changed.

14. Supplementary Table 3: Jan-50 in dilution column

This error has been removed.

15. Were the senior physicians assigning the sepsis/NEC definitions blinded to the other data? Please state.

Yes, two neonatologists, who assigned the sample groupings, were blinded to the immunological data. This has now been stated in the manuscript (line 99) and in Figure 1.

16. 10% FCS is not an ideal stimulation condition, as FCS can substantially activate primary cells such as PBMC. 10% FCS was used throughout all stimulation conditions, so data could be ok for comparisons, but generalizations need to be made very carefully and this limitation should please be acknowledged.

We have now acknowledged this limitation in the text (line 556).

17. Referencing should be expanded to include other important work done in the field.

We have added several new references to the text.

18. There must be an error with some of the birth weights in table 1. Baby A1 surely was not 1780g at birth at 25 weeks. The same question for A7, 1113g at 26 weeks, and B8 and B13 are also doubtful.

Apologies for the mistake in the text. All the birthweights have been double checked with the clinicians and corrected where appropriate.

Reviewer #3 (Remarks to the Author):

This is an excellent paper. The study is well conducted, well written and important.

NEC and sepsis are the leading causes of mortality in babies born at extreme prematurity. Diagnostic discrimination in clinical practice can be challenging and is a major unmet need in implementation of early effective treatment. This is compounded by poor understanding of underpinning disease biology. This study uses state of the art immune profiling techniques (multi-parameter fluorescence cytometry, peripheral blood sc transcriptomics and ELISA) to study NEC and sepsis in this patient group, identifying Amphiregulin as a potential plasma biomarker.

The supplementary data depicts robust flow cytometry gating strategies. The methodology is sound and are provided in enough detail for replication.

There are some potential limitations that should be discussed or clarified if already considered and addressed in the presented dataset:

- The sample size is small, more so once sub-stratified for the various experimental assays (i.e. few babies were phenotyped by more than one assay). However, it is notable how complex it is to sample babies of this prematurity in sufficient volume to then undertake multi-parameter phenotyping.

We thank the reviewer for their comments. We have now included a further 53 samples to our flow cytometry dataset, which represents a 50% increase in the number of total samples in the flow cytometry cohort. Additionally, we have expanded the plasma amphiregulin validation cohort. In this iteration of the manuscript, we include 66 further samples, obtained from seven babies sampled longitudinally, including across sepsis episodes. Therefore, our data are significantly more robust, and indeed, further support that amphiregulin is associated with sepsis and inflammation in preterm babies.

- Linked to above I think many figures amalgamate multiple samples from the same donor. Is it possible that significant findings are driven/weighted by a large number of samples being derived from a single donor in either the healthy or disease cohort? Should matched samples in disease and non-disease groups that originate from the same baby be highlighted in some figures e.g. figure 2 boxplots, and figure 6H for example. I would be interested in the authors' thoughts on this.

Many thanks for this comment. For clarity, we now present a new figure (Fig. 1c) which demonstrates when blood samples were taken from each baby for the flow cytometry study. Critically, we depict clearly which samples represent duplicates (ie. repeat samples from the same sepsis episode) and mark them with a cross (X). These samples are not included in the following comparison between Sepsis and No-Sepsis samples in Fig. 1d-e.

I have one other minor suggestion in that the abstract would be strengthened by defining extreme prematurity

Many thanks for this comment which we have now actioned.

REVIEWER COMMENTS

Reviewer #1 (Remarks to the Author):

Thank you to the authors for performing a significant amount of work for the revisions. However, a number of significant concerns remain.

1. The single cell data remains under analyzed with a very low n.
2. Fig 2c to be clinically relevant would need to compare a pre-sepsis sample to a sepsis sample not to post-sepsis sample.
3. The new proteomics validation data (fig 6) is interesting but rather conflicting and certainly far from demonstrating the ability to use Amph as a biomarker. For example, in patient C5 CRP peaks right around MCS but Amph doesn't peak till much later, for patient C3, there is no Amph peak with MCS, for A13 amph peak is far delayed from CRP, etc.

Reviewer #2 (Remarks to the Author):

The authors have done a lot of work to accommodate the changes suggested by the reviewers. They have adequately addressed most comments; however, some concerns remain.

1. Response to Comment #10

Authors: Whilst we agree that age has an effect on the parameters we measured, we have excluded this as the main driver behind the changes we observed in sepsis (new Extended Data Figure 1 and Figure 2a). Additionally, in response to an earlier reviewer's comment, we have also noted that the male:female ratio is approximately equal within our cohorts.

In my opinion, Extended Data 1 does not convincingly demonstrate that age is not a main driver behind changes observed in sepsis. In fact, it looks more like the opposite; many parameters increase with age. However, there is disparity over time in sample distribution of No sepsis and Sepsis (Fig. 1b); age bracket 1 (Extended Data 1) for example is highly enriched for Sepsis samples whereas age brackets 3-6 are mostly No Sepsis. This effectively indicates that longitudinal assessment of the relationship between postnatal age and y-axis parameters is predominantly dependent on No sepsis samples.

It would be clearer if each age bracket 1-6 (Extended Data 1) was subgrouped into No Sepsis and Sepsis and thus directly assessed within each age bracket for significant differences. In this way, if a reduction to these parameters were to be found in sepsis after effectively pseudo-controlling for age, it would be clearer that these decreases are associated with sepsis. The caveat here is that it is likely that only age brackets 1 and 2 will form effective subgrouping into No sepsis and sepsis, given the rarer sepsis samples in older age groups. Moreover, longitudinal interpretation of the data can then focus on the "more normal" No sepsis samples.

Regarding sepsis data interpretation, it would be advisable to put more emphasis on the changes occurring in sepsis in Fig. 2a rather than Fig. 1d/e, because Fig. 2a is much better temporally controlled, given that Extended Data 1 effectively indicates age should be taken into account.

Of note, in that same comment 10, I suggested a multivariate analysis to assess the data for confounders. The authors did not respond to this aspect of the comment. I still think it is essential to do such an analysis. Identification of confounders (age will almost certainly be one) does not render the current data invalid; rather, discussion of these confounders will substantially improve interpretation of the data and thus the message of the paper.

2. Line 232

It is unclear why CD3+ alpha beta T cells were looked at for DEGs when these cells are not even shown on the UMAP as a possible cell type. Moreover, Extended Data Figure 2 that presumably shows this analysis indicates they are CD3+ T cells.

The decision to look at DEGs in T cells at the CD3 level rather than CD4 and CD8 level is implied to be due to the fourth baby that only had sorted CD3+ T cells. Were the authors not able to determine CD4 and CD8 cell type designations in this 4th infant?

Please clarify cell type identity.

3. Line 323

There is no direct evidence to support impaired functional capacity of NK cells to produce IFN γ and TNF α during sepsis. Particularly as these reductions in sepsis were not observed in the analysis performed in Fig 2a. Please adjust data interpretation.

Reviewer #3 (Remarks to the Author):

I am satisfied by the responses provided by the authors to my comments.

Reviewer #1 (Remarks to the Author):

Thank you to the authors for performing a significant amount of work for the revisions. However, a number of significant concerns remain.

1. The single cell data remains under analyzed with a very low n.

- We acknowledge the reviewer's comment and we have now included a **new** analysis of differential gene expression and pathway analysis for the B cell cluster [NEW Extended Data Figure 3 and manuscript lines 271-293]. These data show that amphiregulin is also amongst the top 50 differentially expressed B cell genes during Enterobacter sepsis and stage III NEC. Additionally, we agree that further analyses might enhance utility of this dataset and have now deposited all files under GEO accession number GSE236099 for readers to explore in greater detail.
- Whilst we agree with the reviewer that increasing the n numbers would have been ideal, we were, unfortunately, not able to do this due to lack of further suitable samples and cessation of recruitment for this study. Nevertheless, we believe our scRNA-seq data remain of significant value as it stands given that no other studies, to our knowledge, have applied scRNA-seq analysis to analyse paired blood samples from extremely preterm babies with NEC/sepsis. Additionally, our initial intention was to use scRNA-seq as a discovery platform which we hope the reviewer agrees was successful; we identified amphiregulin and we then used flow/ELISA to validate this discovery.

Fig 2c to be clinically relevant would need to compare a pre-sepsis sample to a sepsis sample not to post-sepsis sample.

- We fully agree that adding the 'pre' sample would be required to interrogate the utility of immune parameters as potential biomarkers of the early phase sepsis response. Although we do show statistically significant temporal perturbations between pre vs sepsis samples, when all sepsis episodes are analysed together (Fig. 1e), the number of 'pre' samples are significantly diminished when groups are further stratified into CRP static/high groups. Furthermore, because some samples relate to episodes of early-onset sepsis (occurring within 72 hours of birth) we would, by definition, be unable to obtain a 'pre' sample thus excluding these from the analysis.
- In acknowledgement of the reviewer's comment, we have made the following changes:
- A) moved this panel to Extended Data.
- B) Shifted our entire emphasis in the text away from immune parameters as early biomarkers. Instead, we highlight evaluating their utility as adjunctive datapoints a clinician might use, alongside standard tests, to 'rule-out' sepsis and guide antibiotic de-escalation. This is of particular importance to neonatal paediatricians. Immuno-

infective disorders (invasive sepsis and necrotising enterocolitis) are now listed among the leading causes of death in babies born before 27 weeks gestation. Prolonged courses of antibiotics in this patient population have been shown to be an independent risk factor for both septicaemia and NEC and thus, we feel that this latter analysis and new focus of the manuscript is not reliant upon pre vs sepsis changes whilst at the same time addressing an area of unmet clinical need.

- 3. The new proteomics validation data (fig 6) is interesting but rather conflicting and certainly far from demonstrating the ability to use Amph as a biomarker. For example, in patient C5 CRP peaks right around MCS but Amph doesn't peak till much later, for patient C3, there is no Amph peak with MCS, for A13 amph peak is far delayed from CRP, etc.

We have taken this point on board and have made the following adjustments:

- 1) We have increased the size of our validation cohort considerably: Our clinical collaborators reviewed the request to expand the cohort and took the view that the research findings were important to validate. They provided access to a further 91 samples from 11 babies but have asked that serious consideration is given by the journal and reviewers to acknowledge the nature of these precious and difficult to obtain plasma samples taken from a cohort of extremely preterm babies. The additional samples increase the number of total plasma samples in our validation cohort from **39** (from **5** longitudinally sampled **babies** from one hospital cohort) in the first iteration of the manuscript to **224 samples** (from **23 babies** across two independent hospital cohorts). In addition, we now also include analysis of plasma amphiregulin from 9 preterm babies (n=23 samples) who did not encounter sepsis or NEC as a comparison (NEW Fig. 6a). Addition of these data further strengthen the observation that only amphiregulin, but not other immune analytes commonly associated with neonatal sepsis (e.g. IL-6), can differentiate Sepsis samples with low CRP, from No-Sepsis samples (Fig. 6d).
- 2) We agree with the reviewer that amphiregulin is not raised during the early phase of sepsis in all cases. We would perhaps not expect a perfect correlation in such a diverse cohort of patients and no biomarker is perfect. Nevertheless, as the reviewer pointed out, in some episodes amphiregulin is raised but not CRP, whilst in other instances, CRP is raised before amphiregulin, or in the absence of amphiregulin. We have now directly pointed this out in the text (Lines 334-340).
- Additionally, we have shifted emphasis away from 'early phase biomarkers' to now discuss the potential utility of combining amphiregulin with CRP to 'rule-out' rather than 'rule-in' sepsis. In a new analysis, we show that a test which combines CRP and amphiregulin together, has higher sensitivity and negative predictive value (**Sensitivity 0.97, NPV 0.94**) compared to a test based on CRP alone (**Sensitivity 0.55, NPV 0.65**), indicating a greater potential ability to rule-out sepsis. We qualify in the text that such tests require prospective evaluation in larger validation cohorts (NEW Fig. 6e and NEW Extended Data Fig. 6 and manuscript lines 350-371).

You could argue that such tests are in great need to help clinicians more confidently stop unnecessary antibiotics if sepsis could be ruled out with greater certainty. Indeed, the utility of this may be greater than that of a biomarker. It is well established that many more babies are started on antibiotics than have proven bacterial infections. As previously alluded to, prolonged antibiotic exposure in this group drives gut dysbiosis leading to further episodes of NEC and sepsis, whilst additionally increasing the likelihood of antimicrobial resistance, one of the top threats to neonatal health today. Thus, even modest reductions in antibiotic use guided by such tests might prove of significant benefit to reducing rates of sepsis, NEC, days in hospital and onward risk of antimicrobial resistant infections.

- 3) Finally, we would like to present here further data for the reviewers only, on the potential importance of measuring amphiregulin from a prognostic point of view.
 - In our cohort, we had a low mortality rate (1/30 babies), therefore inferences as to the utility of plasma amphiregulin as a prognostic indicator of disease severity/mortality could not be made. In order to understand these relationships further, we turned to publicly available whole blood transcriptional datasets, obtained from paediatric septic shock patients across two studies. Strikingly, we observed that AREG was the top upregulated gene (GSE4607), or within the top 6 upregulated genes (GSE26378), in sepsis non-survivors versus survivors. We would be happy to include these data as an extended data panel should the reviewers advise.
 - These data further support our hypothesis that measurement of amphiregulin during neonatal sepsis is likely to be important and in cases where it is high, might add further justification to continue antibiotics.

Reviewer Figure 1. Whole blood transcriptional profiling, comparing non-survivors versus survivors in two paediatric septic shock cohorts. Volcano plots show log fold change (logfc) versus $-\log_{10}$ P value. AREG is annotated on each plot.

Reviewer #2 (Remarks to the Author):

The authors have done a lot of work to accommodate the changes suggested by the reviewers. They have adequately addressed most comments; however, some concerns remain.

1. Response to Comment #10

Authors: Whilst we agree that age has an effect on the parameters we measured, we have excluded this as the main driver behind the changes we observed in sepsis (new Extended Data Figure 1 and Figure 2a). Additionally, in response to an earlier reviewer's comment, we have also noted that the male:female ratio is approximately equal within our cohorts.

In my opinion, Extended Data 1 does not convincingly demonstrate that age is not a main driver behind changes observed in sepsis. In fact, it looks more like the opposite; many parameters increase with age. However, there is disparity over time in sample distribution of No sepsis and Sepsis (Fig. 1b); age bracket 1 (Extended Data 1) for example is highly enriched for Sepsis samples whereas age brackets 3-6 are mostly No Sepsis. This effectively indicates that longitudinal assessment of the relationship between postnatal age and y-axis parameters is predominantly dependent on No sepsis samples.

It would be clearer if each age bracket 1-6 (Extended Data 1) was subgrouped into No Sepsis and Sepsis and thus directly assessed within each age bracket for significant differences. In this way, if a reduction to these parameters were to be found in sepsis after effectively pseudo-controlling for age, it would be clearer that these decreases are associated with sepsis. The caveat here is that it is likely that only age brackets 1 and 2 will form effective subgrouping into No sepsis and sepsis, given the rarer sepsis samples in older age groups. Moreover, longitudinal interpretation of the data can then focus on the "more normal" No sepsis samples.

We thank the reviewer for their comments which we have tried to address as fully as possible in the following manor:

- As suggested by the reviewer, we have re-plotted our longitudinal graphs with 'No-Sepsis' samples only, which now more accurately depict age-associated changes due to immune development (NEW Extended Data Fig. 1a and manuscript lines 142-150).
- We acknowledge the reviewers comment about 'disparity over time in sample distribution' and that age brackets '3-6 are mostly No Sepsis'. To address this, we have sub-grouped age brackets 1-2 together and brackets 3-6 together and can now show that most sepsis-induced immune perturbations are preserved in both age strata (Extended Data Fig. 1b and manuscript lines 151-158).

Regarding sepsis data interpretation, it would be advisable to put more emphasis on the changes occurring in sepsis in Fig. 2a rather than Fig. 1d/e, because Fig. 2a is much better

temporally controlled, given that Extended Data 1 effectively indicates age should be taken into account.

- We have taken this on board and have now removed Fig. 1e and replaced this with the previous Fig. 2a so that the temporal changes are more prominent.

Of note, in that same comment 10, I suggested a multivariate analysis to assess the data for confounders. The authors did not respond to this aspect of the comment. I still think it is essential to do such an analysis. Identification of confounders (age will almost certainly be one) does not render the current data invalid; rather, discussion of these confounders will substantially improve interpretation of the data and thus the message of the paper.

- We agree with the reviewer that multivariate analyses would certainly be a powerful method to assess for confounders in these data. We have discussed this with epidemiology/statistics colleagues who run these models frequently, and their advice was that our analysis would be underpowered with our current sample sizes.
- However, for the benefit of the reviewers, we present a multivariate analysis below, which assesses the influence of sepsis on several of the most significant immune parameters we have identified (e.g. CD8⁺ T cell count), whilst controlling for the following confounders: postnatal age, gestational age, sex, birth weight Z-score, respiratory support and antenatal steroid exposure.
- These data demonstrate that sepsis-induced perturbations in CD8⁺ T cell numbers, classical monocyte median HLA-DR, mDC, pDC and T cell frequencies and plasma amphiregulin levels are preserved after adjustment for confounders. Additionally, these data further acknowledge the importance of factoring in postnatal age into analyses of immune perturbations in early life, as the reviewer has mentioned.
- We have added a new section to the manuscript to discuss the potential influence of confounders (Manuscript lines 452-467).

Dependent variable		% T cells				
Fixed effects	Estimate	Lower 95% CI	Upper 95% CI	Std Error	P val	
Intercept	31.99419	-108.0443	172.03268	62.85231	0.62176554	
Sepsis- Yes	-8.53968	-12.53865	-4.540708	2.018843	0.000047	
Postnatal age	2.906479	1.8011131	4.0118443	0.558098	0.0000008	
Sex (Female)	-5.811171	-17.68533	6.0629861	5.428653	0.30619056	
Gestational age at birth	0.034728	-0.761188	0.8306445	0.356436	0.924339052	
Birth weight Z-score	-1.602624	-8.856251	5.6510026	3.256415	0.633219814	
Respiratory Support	8.919838	-2.356141	20.195817	5.183401	0.110574675	
Antenatal Steroids	-5.437445	-15.21687	4.3419789	4.486615	0.248946646	

Dependent variable		CD8+ T cells/ml blood				
Fixed effects	Estimate	Lower 95% CI	Upper 95% CI	Std Error	P val	
Intercept	-35573.66	-371879.4	300732.1	154113.9	0.821391273	
Sepsis- Yes	-11906.22	-19999.32	-3813.114	4084.327	0.004	
Postnatal age	5735.897	3553.5024	7918.2909	1101.445	0.00000087	
Sex (Female)	-8136.104	-36227.4	19955.189	13065.35	0.543734912	
Gestational age at birth	392.1201	-1520.702	2304.9426	875.4187	0.662390257	
Birth weight Z-score	5059.605	-12382.6	22501.806	7990.74	0.53868651	
Respiratory Support	10623.21	-16088.98	37335.394	12437.27	0.407612135	
Antenatal Steroids	-5210.159	-28276.87	17856.548	10757.23	0.635615875	

Dependent variable		% mDC				
Fixed effects	Estimate	Lower 95% CI	Upper 95% CI	Std Error	P val	
Intercept	-16.64747	-38.45475	5.1598231	9.99927	0.122030189	
Sepsis- Yes	-1.114541	-1.822114	-0.406968	0.357274	0.0023	
Postnatal age	0.437371	0.2419628	0.6327781	0.098678	0.00002	
Sex (Female)	-0.548563	-2.418445	1.3213198	0.869971	0.538717635	
Gestational age at birth	0.09671	-0.027105	0.2205237	0.056653	0.114232787	
Birth weight Z-score	0.239291	-0.890068	1.3686495	0.5182	0.652522962	
Respiratory Support	-0.202745	-1.98236	1.5768713	0.833343	0.81114731	
Antenatal Steroids	-0.757701	-2.301757	0.7863557	0.720312	0.310557293	

Dependent variable		%pDC				
Fixed effects	Estimate	Lower 95% CI	Upper 95% CI	Std Error	P val	
Intercept	-28.60106	-57.90302	0.7008962	13.43938	0.054871969	
Sepsis- Yes	-0.966306	-1.7575	-0.175111	0.399461	0.017	
Postnatal age	0.381818	0.163079	0.6005572	0.110447	0.00076	
Sex (Female)	-0.324194	-2.798619	2.1502306	1.151996	0.782569222	
Gestational age at birth	0.164344	-0.002202	0.3308905	0.076265	0.05263378	
Birth weight Z-score	0.264264	-1.254275	1.7828031	0.696446	0.711029122	
Respiratory Support	0.172495	-2.18045	2.5254402	1.099002	0.877478676	
Antenatal Steroids	-0.349756	-2.385737	1.6862251	0.950538	0.718332307	

Dependent variable		Classical monocytes median HLA-DR				
Fixed effects	Estimate	Lower 95% CI	Upper 95% CI	Std Error	P val	
Intercept	-21307.88	-71576.67	28960.914	23190.32	0.375463359	
Sepsis- Yes	-2384.799	-4431.91	-337.6883	1032.498	0.023	
Postnatal age	1634.868	1068.7536	2200.983	285.5665	0.00000009	
Sex (Female)	2557.196	-1801.47	6915.8616	2034.215	0.229076789	
Gestational age at birth	163.4594	-121.5722	448.49098	131.1628	0.235909928	
Birth weight Z-score	1441.972	-1150.068	4034.0108	1193.7	0.24962455	
Respiratory Support	-1862.083	-6003.42	2279.2539	1939.217	0.352497779	
Antenatal Steroids	-2.354605	-3623.116	3618.4073	1695.975	0.998910892	

Dependent variable		Amphiregulin pg/ml				
Fixed effects	Estimate	Lower 95% CI	Upper 95% CI	Std Error	P val	
Intercept	257.9517	-328.0711	843.97457	267.0034	0.3543	
Sepsis- Yes	22.60128	0.6423063	44.56025	11.107	0.04	
Postnatal age	-15.33985	-23.33515	-7.344551	4.047324	0.0002	
Sex (Female)	-20.55683	-52.16165	11.047989	14.44363	0.1811	
Gestational age at birth	-0.591877	-3.889708	2.7059534	1.495656	0.7	
Birth weight Z-score	-7.494781	-25.28903	10.299468	8.010502	0.371	

Multivariate analysis to assess for confounders. Data were analysed by a linear mixed-effects model fit by REML. Baby ID was set as a random effect to account for repeated measures.

Flow cytometry parameters; Sepsis n=32, No-Sepsis n=104 (see Fig. 1c).

Plasma amphiregulin; Sepsis n=47, No-Sepsis n=116 (see Fig. 6b).

Dependent variables: % T cells, CD8⁺ T cell numbers, %mDC, %pDC, classical monocyte median HLA-DR and plasma amphiregulin.

Fixed effects: Sepsis (Yes); postnatal age; female sex; gestational age; birthweight Z-score; respiratory support; antenatal steroids.

Note that the cohorts used for flow cytometry and amphiregulin analysis were not entirely overlapping as shown in Table 1. Respiratory support and antenatal steroids were not shown for amphiregulin due to missing values.

2. Line 232

It is unclear why CD3⁺ alpha beta T cells were looked at for DEGs when these cells are not even shown on the UMAP as a possible cell type. Moreover, Extended Data Figure 2 that presumably shows this analysis indicates they are CD3⁺ T cells.

The decision to look at DEGs in T cells at the CD3 level rather than CD4 and CD8 level is implied to be due to the fourth baby that only had sorted CD3⁺ T cells. Were the authors not able to determine CD4 and CD8 cell type designations in this 4th infant?

Please clarify cell type identity.

We apologize for the lack of clarity in our wording and illustration on the UMAP. When we mentioned CD3⁺ $\alpha\beta$ -T cells, we were referring to CD4⁺ T cells and CD8⁺ T cells together, although we now realise that we did not make this clear in the text. We have therefore updated the UMAPs to illustrate this (Fig. 2b and Supplementary Fig. 8). We made this decision given that there were very few differences in differential gene expression between CD4⁺ and CD8⁺ T cell clusters separately.

This was also the case for the 4th infant in whom we were able to identify both cell types, though chose to combine them and analyse total CD3⁺ $\alpha\beta$ -T cells for the same reason as stated above.

3. Line 323

There is no direct evidence to support impaired functional capacity of NK cells to produce IFN γ and TNF α during sepsis. Particularly as these reductions in sepsis were not observed in the analysis performed in Fig 2a. Please adjust data interpretation.

We acknowledge the lack of temporal data to show changes in these parameters and we have thus removed these data from our analyses (though they remain in our source data for readers to access and interpret).

Reviewer #3 (Remarks to the Author):

I am satisfied by the responses provided by the authors to my comments.

Many thanks.

REVIEWERS' COMMENTS

Reviewer #2 (Remarks to the Author):

The authors have addressed my additional comments mostly satisfactorily.

One remaining point re multivariate analysis (MVA): I understand - and in fact expected - that the study is underpowered to conduct a complete and statistically rock-solid MVA that resolves all confounder issues.

Nonetheless, I would strongly recommend that the authors include the MVA they have conducted in the final manuscript. They can explain the circumstances and statistics, expanding the paragraph they have already added (line 452 onwards). As I said, it is important for readers to understand the possible influence of confounders, the authors' attempt to describe them and control for them, the statistical challenges of doing so, and the results of this analysis.

Including the MVA will strengthen the paper's messages, despite the statistical challenges.

Point-by-point response to REVIEWERS' COMMENTS

Reviewer #2 (Remarks to the Author):

The authors have addressed my additional comments mostly satisfactorily.

One remaining point re multivariate analysis (MVA): I understand - and in fact expected - that the study is underpowered to conduct a complete and statistically rock-solid MVA that resolves all confounder issues.

Nonetheless, I would strongly recommend that the authors include the MVA they have conducted in the final manuscript. They can explain the circumstances and statistics, expanding the paragraph they have already added (line 452 onwards). As I said, it is important for readers to understand the possible influence of confounders, the authors' attempt to describe them and control for them, the statistical challenges of doing so, and the results of this analysis.

Including the MVA will strengthen the paper's messages, despite the statistical challenges.

We thank the reviewer for their comment, and we have now included the multivariate analysis within the manuscript (Supplementary Figure 7). We have additionally expanded the paragraph, beginning line 456, as advised.